# Implicit variance regularization in non-contrastive SSL

**Manu Srinath Halvagal**[1,2,*]       **Axel Laborieux**[1,*]       **Friedemann Zenke**[1,2]

`{firstname.lastname}@fmi.ch`
[1] Friedrich Miescher Institute for Biomedical Research, Basel, Switzerland
[2] Faculty of Science, University of Basel, Basel, Switzerland
[*] These authors contributed equally.

## Abstract

Non-contrastive self-supervised learning (SSL) methods like BYOL and SimSiam rely on asymmetric predictor networks to avoid representational collapse without negative samples. Yet, how predictor networks facilitate stable learning is not fully understood. While previous theoretical analyses assumed Euclidean losses, most practical implementations rely on cosine similarity. To gain further theoretical insight into non-contrastive SSL, we analytically study learning dynamics in conjunction with Euclidean and cosine similarity in the eigenspace of closed-form linear predictor networks. We show that both avoid collapse through implicit variance regularization albeit through different dynamical mechanisms. Moreover, we find that the eigenvalues act as effective learning rate multipliers and propose a family of isotropic loss functions (IsoLoss) that equalize convergence rates across eigenmodes. Empirically, IsoLoss speeds up the initial learning dynamics and increases robustness, thereby allowing us to dispense with the exponential moving average (EMA) target network typically used with non-contrastive methods. Our analysis sheds light on the variance regularization mechanisms of non-contrastive SSL and lays the theoretical grounds for crafting novel loss functions that shape the learning dynamics of the predictor's spectrum.

## 1 Introduction

SSL has emerged as a powerful method to learn useful representations from vast quantities of unlabeled data [1–8]. In SSL, the network's objective is to "pull" together its outputs for two differently augmented versions of the same input, so that they learn representations that are predictive across randomized transformations [9]. To avoid the trivial solution whereby the network output becomes constant, also called representational collapse, SSL methods use either a contrastive objective to "push" apart representations of unrelated images [2, 3, 10–13] or other non-contrastive strategies. Non-contrastive methods comprise explicit variance regularization techniques [6, 7, 14], whitening approaches [15, 16], and asymmetric losses as in Bootstrap Your Own Latent (BYOL) [1] and SimSiam [5]. Asymmetric losses break symmetry between the two branches by passing one of the representations through a predictor network and stopping gradients from flowing through the other "target" branch. How this architectural modification prevents representational collapse is not obvious and has been the focus of several theoretical [17–21] and empirical studies [22–24]. A significant advance was provided by Tian et al. [17] who showed that linear predictors align with the correlation matrix of the embeddings, and proposed the closed-form predictor DirectPred based on this insight. However, previous analyses assumed a Euclidean loss at the output [17–19, 21] except [20], whereas practical implementations typically use the cosine loss [1, 5] which yields superior performance on downstream tasks. This difference raises the question whether analysis based on the Euclidean loss provides an accurate account of the learning dynamics under the cosine loss.

37th Conference on Neural Information Processing Systems (NeurIPS 2023).

In this work, we provide a comparative analysis of the learning dynamics for the Euclidean and cosine-based asymmetric losses in the eigenspace of the closed-form predictor DirectPred. Our analysis shows how both losses implicitly regularize the variance of the representations, revealing a connection between asymmetric losses and explicit variance regularization in VICReg [7]. Yet, the learning dynamics induced by the two losses are markedly different. While the learning dynamics of different eigenmodes decouple in the Euclidean case, dynamics remain coupled for the cosine loss.

Moreover, our analysis shows that for both losses, the predictor's eigenvalues act as learning rate multipliers, thereby slowing down learning for modes with small eigenvalues. Based on our analysis, we craft an isotropic loss function (IsoLoss) for each case that resolves this problem and speeds up the initial learning dynamics. Furthermore, IsoLoss works without an EMA target network possibly because it boosts small eigenvalues, the purported role of the EMA in DirectPred [17]. In summary, our main contributions are the following:

- We analyze the SSL dynamics in the eigenspace of closed-form linear predictors for asymmetric Euclidean and cosine losses and show that both perform implicit variance regularization, but with markedly different learning dynamics.

- Our analysis shows that predictor eigenvalues act as learning rate multipliers which slows down learning for small eigenvalues.

- We propose isotropic loss functions for both cases that equalize the dynamics across eigenmodes and improve robustness, thereby allowing to learn without an EMA target network.

## 2 Eigenspace analysis of the learning dynamics

To gain a better analytic understanding of the SSL dynamics underlying non-contrastive methods such as BYOL and SimSiam [1, 5], we analyze them in the predictor's eigenspace. Specifically we proceed in three steps. First, building on DirectPred, we invoke the neural tangent kernel (NTK) to derive simple dynamic expressions of the predictor's eigenmodes for Euclidean and cosine loss. This formulation uncovers the implicit variance regularization mechanisms that prevent representational collapse. Using the eigenspace framework, we illustrate how removing the predictor or the stop-gradient results in collapse or run-away dynamics. Finally, we find that predictor eigenvalues act as learning rate multipliers for their associated mode, thereby slowing down learning for small eigenvalues. We derive a modified isotropic loss function (IsoLoss) that provides more equalized learning dynamics across modes, which showcases how our analytic insights help to design novel loss functions that actively shape the predictor spectrum. However, before we start our analysis, we will briefly review DirectPred [17] and the NTK [25], a powerful theoretical tool linking representational changes and parameter updates. We will rely on both concepts for our analysis.

### 2.1 Background and problem setup

We begin by reviewing DirectPred [17] and defining our notation. In the following, we consider a Siamese neural network $z = f(x; \theta)$ with output $z \in \mathbb{R}^M$, input $x \in \mathbb{R}^N$, and parameters $\theta$. We further assume a linear predictor network $W_{\mathrm{P}} \in \mathbb{R}^{M \times M}$ and use the same parameters for the online and target branches as in SimSiam [5]. We denote pairs of representations as $z^{(1)}, z^{(2)}$ corresponding to pairs of inputs $x^{(1)}, x^{(2)}$ related through augmentation and implicitly assume that all losses are averaged over many augmented pairs. The asymmetric loss function (Fig. 1a), introduced in BYOL [1], is then given by:

$$\mathcal{L} = d\left(W_{\mathrm{P}} z^{(1)}, \mathrm{SG}(z^{(2)})\right),$$

where SG denotes the stop-gradient operation, and $d$ is either the Euclidean distance metric $d(a, b) = \frac{1}{2}\|a - b\|^2$ or the cosine distance metric $d(a, b) = -\frac{a^\top b}{\|a\|\|b\|}$. We refer to the corresponding loss functions as $\mathcal{L}^{\mathrm{euc}}$ and $\mathcal{L}^{\mathrm{cos}}$ respectively.

**DirectPred.** Tian et al. [17] showed that a linear predictor in the BYOL setting aligns during learning with the correlation matrix of representations $C_z := \mathbb{E}_x\left[z z^\top\right]$, where the expectation is taken over the data distribution. Since the correlation matrix is a real symmetric matrix, one can diagonalize it over $\mathbb{R}$: $C_z = U D_C U^\top$, where $U$ is an orthogonal matrix whose columns are the eigenvectors of $C_z$ and $D_C$ is the real-valued diagonal matrix of the eigenvalues $s_m$ with $m \in [1, M]$.

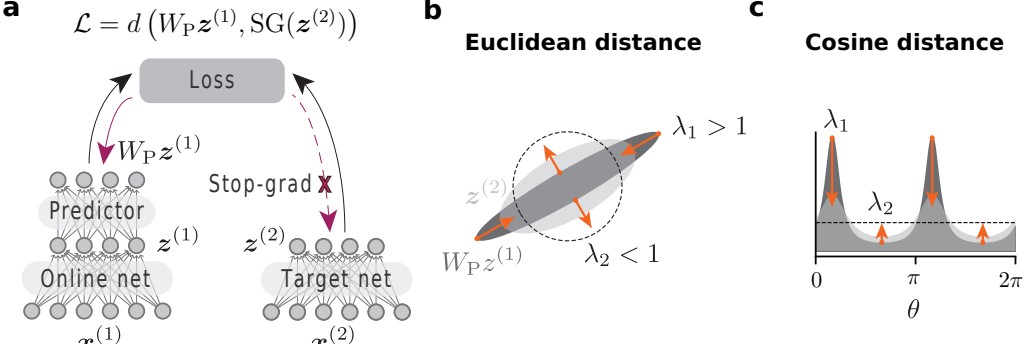

Figure 1: **(a)** Schematic of a Siamese network with a predictor network and a stop-gradient on the target network branch. The target network can be a copy (SimSiam [5]) or a moving average (BYOL [1]) of the online network. In either case, the target network is not optimized with gradient descent. **(b)** Visualization of learning dynamics under the Euclidean distance metric showing learning update directions along two eigenmodes, with the light cloud representing the distribution of the representations $z$, the darker cloud representing the predictor outputs $W_P z$, and the dotted circle indicates the steady state $\lambda_{1,2} = 1$, reached during learning. All eigenvalues converge to one. **(c)** Same as **(b)**, but for the cosine distance. The dotted line indicates the steady state $\lambda_1 = \lambda_2$.

Given this eigendecomposition, the authors proposed DirectPred, in which the predictor is not learned via gradient descent but directly set to:

$$W_P = f_\alpha\left(C_z\right) = U D_C^\alpha U^\top \quad , \tag{1}$$

where $\alpha$ is a positive constant exponent applied element-wise to $D_C$. The eigenvalues $\lambda_m$ of the predictor matrix $W_P$ are then $\lambda_m = s_m^\alpha$. We use $D$ to denote the diagonal matrix containing the eigenvalues $\lambda_m$. While DirectPred used $\alpha = 0.5$, the follow-up study DirectCopy [18] showed that $\alpha = 1$ is also effective while avoiding the expensive diagonalization step. While Tian et al. [17] based their analysis on the Euclidean loss $\mathcal{L}^{\mathrm{euc}}$, most practical models, including Tian et al.'s large-scale experiments, relied on the cosine similarity loss $\mathcal{L}^{\mathrm{cos}}$. This discrepancy raises the question to what extent setting the predictor to the above expression is justified for the cosine loss. Empirically, we find that a trainable linear predictor *does* align its eigenspace with that of the representation correlation matrix also for the cosine loss (see Fig. 4 in Appendix A).

**Neural tangent kernel (NTK).** The NTK is a powerful analytical tool characterizing the learning dynamics of neural networks [25, 26]. Here, we recall the definition of the *empirical* NTK [26] corresponding to a single instantiation of the network's parameters $\boldsymbol{\theta}$. If $|\mathsf{D}|$ denotes the size of the training dataset, $\mathcal{L}: \mathbb{R}^M \to \mathbb{R}$ an arbitrary loss function, $\mathcal{X}$, the training data concatenated into one vector of size $N|\mathsf{D}|$, and $\mathcal{Z} = z(\mathcal{X})$, the concatenated output of size $M|\mathsf{D}|$, then the empirical NTK is the $(M|\mathsf{D}| \times M|\mathsf{D}|)$-sized matrix:

$$\Theta_t(\mathcal{X}, \mathcal{X}) = \nabla_{\boldsymbol{\theta}} \mathcal{Z} \nabla_{\boldsymbol{\theta}} \mathcal{Z}^\top,$$

and the continuous-time gradient-descent dynamics [26] of the representations $z$ are given by:

$$\frac{\mathrm{d}z}{\mathrm{d}t} = -\eta \Theta_t(\boldsymbol{x}, \mathcal{X}) \nabla_{\mathcal{Z}} \mathcal{L} \quad . \tag{2}$$

In other words, the empirical NTK links the representational dynamics $\frac{\mathrm{d}z}{\mathrm{d}t}$ under gradient descent on the parameters $\boldsymbol{\theta}$, and the "representational gradient" $\nabla_{\mathcal{Z}} \mathcal{L}$.

## 2.2 Implicit variance regularization in non-contrastive SSL

As a starting point for our analysis, we first express the relevant loss functions in the eigenbasis of the predictor network. We do this using a closed-form linear predictor as prescribed by DirectPred. In the following, we use $\hat{z} = U^\top z$ to denote the representation expressed in the eigenbasis.

**Lemma 1.** (Euclidean and cosine loss in the predictor eigenspace) *Let $W_{\mathrm{P}}$ be a linear predictor set according to DirectPred with eigenvalues $\lambda_m$, and $\hat{z}$ the representations expressed in the predictor's eigenbasis. Then the asymmetric Euclidean loss $\mathcal{L}^{\mathrm{euc}}$ and cosine loss $\mathcal{L}^{\mathrm{cos}}$ can be expressed as:*

$$\mathcal{L}^{\mathrm{euc}} = \tfrac{1}{2}\sum_m^M |\lambda_m \hat{z}_m^{(1)} - \mathrm{SG}(\hat{z}_m^{(2)})|^2 \quad, \tag{3}$$

$$\mathcal{L}^{\mathrm{cos}} = -\sum_m^M \frac{\lambda_m \hat{z}_m^{(1)}\mathrm{SG}(\hat{z}_m^{(2)})}{\|D\hat{\boldsymbol{z}}^{(1)}\|\|\mathrm{SG}(\hat{\boldsymbol{z}}^{(2)})\|} \quad. \tag{4}$$

for which we defer the simple proof to Appendix B. Rewriting the losses in the eigenbasis makes it clear that the asymmetric loss with DirectPred can be viewed as an *implicit* loss function in the predictor's eigenspace, where the variance of each mode naturally appears through the $\lambda_m$ terms. In the following analysis, we will show how the learning dynamics implicitly regularize these variances $\lambda_m$. From Eq. (3) we directly see that $\mathcal{L}^{\mathrm{euc}}$ is a sum of $M$ terms, one for each eigenmode, which decouples the learning dynamics, a fact first noted by Tian et al. [17]. In contrast, the form of $\mathcal{L}^{\mathrm{cos}}$ yields coupled dynamics due to the $\|D\hat{\boldsymbol{z}}^{(1)}\| = \sqrt{\sum_k(\lambda_k\hat{z}_k^{(1)})^2}$ term in the denominator. This coupling arises from the normalization of the representation vectors to the unit hypersphere when calculating the cosine distance. The normalization effectively removes one degree of freedom and, in the process, adds a dependence between all the representation dimensions (Fig. 1b and 1c).

To get an analytic handle on the evolution of the eigen-representations $\hat{z}$ as the encoder learns, we first note that if training were to update the representations directly, instead of indirectly through updating the weights $\boldsymbol{\theta}$, they would evolve along the following "representational gradients":

$$\nabla_{\hat{\boldsymbol{z}}^{(1)}}\mathcal{L}^{\mathrm{euc}} = \left(D\hat{\boldsymbol{z}}^{(1)} - \hat{\boldsymbol{z}}^{(2)}\right)D \quad, \tag{5}$$

$$\nabla_{\hat{\boldsymbol{z}}^{(1)}}\mathcal{L}^{\mathrm{cos}} = -\frac{D\hat{\boldsymbol{z}}^{(2)}}{\|D\hat{\boldsymbol{z}}^{(1)}\|\|\hat{\boldsymbol{z}}^{(2)}\|} + \frac{(D\hat{\boldsymbol{z}}^{(1)})^\top\hat{\boldsymbol{z}}^{(2)}}{\|D\hat{\boldsymbol{z}}^{(1)}\|^3\|\hat{\boldsymbol{z}}^{(2)}\|}D^2\hat{\boldsymbol{z}}^{(1)} \quad. \tag{6}$$

In practice, however, representations of different samples do not evolve independently along these gradients, but influence each other through parameter changes in $\boldsymbol{\theta}$. This interdependence of representations and parameters are captured by the empirical NTK $\Theta_t(\mathcal{X}, \mathcal{X})$ (cf. Eq. (2)). Because the NTK is positive semi-definite, loosely speaking, gradient descent on the parameters changes representations "in the direction" of the above representational gradients.

To see this link more formally, we express the NTK in the eigenbasis as $\hat{\Theta}_t(\mathcal{X},\mathcal{X}) = \nabla_{\boldsymbol{\theta}}\hat{\mathcal{Z}}\nabla_{\boldsymbol{\theta}}\hat{\mathcal{Z}}^\top$ where $\hat{\mathcal{Z}} = \hat{z}_t(\mathcal{X}) = U^\top z_t(\mathcal{X})$. Since we are concerned with the learning dynamics in this rotated basis, we will rewrite Eq. (2) for continuous-time gradient descent for a generic loss function $\mathcal{L}$ as:

$$\frac{\mathrm{d}\hat{\boldsymbol{z}}}{\mathrm{d}t} = -\eta\hat{\Theta}_t(\boldsymbol{x},\mathcal{X})\nabla_{\hat{\mathcal{Z}}}\mathcal{L} \quad. \tag{7}$$

Note, that structurally these dynamics are the same as the embedding space dynamics in Eq. (2) but merely expressed in the predictor eigenbasis (see Lemma 2 in Appendix B for a derivation). Although $\hat{\Theta}_t$ changes over time and is generally intractable in finite-width networks, it is positive semidefinite. This property guarantees that the cosine angle between the representational training dynamics under the parameter-space optimization of a neural network $\frac{\mathrm{d}}{\mathrm{d}t}\hat{\mathcal{Z}} \propto -\hat{\Theta}_t\nabla_{\hat{\mathcal{Z}}}\mathcal{L}$ and the dynamics that would result from optimizing the representations $\frac{\mathrm{d}}{\mathrm{d}t}\hat{\mathcal{Z}} \propto -\nabla_{\hat{\mathcal{Z}}}\mathcal{L}$ is non-negative:

$$\left\langle -\nabla_{\hat{\mathcal{Z}}}\mathcal{L}, \frac{\mathrm{d}\hat{\mathcal{Z}}}{\mathrm{d}t}\right\rangle = \eta\left\langle\nabla_{\hat{\mathcal{Z}}}\mathcal{L}, \hat{\Theta}_t\nabla_{\hat{\mathcal{Z}}}\mathcal{L}\right\rangle \geq 0.$$

In other words, the representational updates due to network training lie within a 180-degree cone of the dynamics prescribed by Eqs. (5) and (6). This guarantee makes it possible to draw qualitative conclusions about asymptotic collective behavior, e.g., whether a network is bound to collapse or not, from analyzing the more tractable dynamics that follow the representational gradients $\frac{\mathrm{d}}{\mathrm{d}t}\hat{\mathcal{Z}} \propto -\nabla_{\hat{\mathcal{Z}}}\mathcal{L}$ of the transformed BYOL/SimSiam loss. For ease of analysis, we now consider linear networks with Gaussian i.i.d inputs, an important limiting case amenable for theoretical analysis [27]. In this setting

the empirical NTK becomes the identity and the simplified representational dynamics are exact, allowing us to fully characterize the representational dynamics for $\mathcal{L}^{\text{euc}}$ and $\mathcal{L}^{\text{cos}}$ in the following two theorems. In the proofs for these theorems, we show that the assumption of Gaussian inputs can be relaxed further.

**Theorem 1.** (Representational dynamics under $\mathcal{L}^{\text{euc}}$) *For a linear network with i.i.d Gaussian inputs learning with $\mathcal{L}^{\text{euc}}$, the representational dynamics of each mode $m$ independently follow the gradient of the loss $-\nabla_{\hat{z}}\mathcal{L}^{\text{euc}}$. More specifically, the dynamics uncouple and follow $M$ independent differential equations:*

$$\frac{\mathrm{d}\hat{z}_m^{(1)}}{\mathrm{d}t} = -\eta\frac{\partial\mathcal{L}^{\text{euc}}}{\partial\hat{z}_m^{(1)}}(t) = \eta\lambda_m\left(\hat{z}_m^{(2)} - \lambda_m\hat{z}_m^{(1)}\right) \quad, \tag{8}$$

*which, after taking the expectation over augmentations yields the dynamics:*

$$\frac{\mathrm{d}\hat{z}_m}{\mathrm{d}t} = \eta\lambda_m\left(1 - \lambda_m\right)\hat{z}_m \quad. \tag{9}$$

We provide the proof in Appendix B and appreciate that $\frac{\mathrm{d}}{\mathrm{d}t}\hat{z}_m$ has the same sign as $\hat{z}_m$ whenever $\lambda_m < 1$ and the opposite sign whenever $\lambda_m > 1$. These dynamics are convergent and approach an eigenvalue $\lambda_m$ of one, thereby preventing collapse of mode $m$. Since the eigenmodes are orthogonal and uncorrelated, and the condition simultaneously holds for all modes, this ultimately prevents both representational and dimensional collapse [28]. Since the eigenvalues also correspond to the variance of the representations, the underlying mechanism constitutes an *implicit* form of variance regularization. Finally, we note that the above decoupling of the dynamics for the Euclidean loss has been described previously in Tian et al. [17].

Nevertheless, the representational dynamics are different for the commonly used cosine loss $\mathcal{L}^{\text{cos}}$.

**Theorem 2.** (Representational dynamics under $\mathcal{L}^{\text{cos}}$) *For a linear network with i.i.d Gaussian inputs trained with $\mathcal{L}^{\text{cos}}$, the dynamics follow a system of $M$ coupled differential equations:*

$$\frac{\mathrm{d}\hat{z}_m^{(1)}}{\mathrm{d}t} = \eta\frac{\lambda_m}{\|D\hat{z}^{(1)}\|^3\|\hat{z}^{(2)}\|}\sum_{k\neq m}\lambda_k\left(\lambda_k(\hat{z}_k^{(1)})^2\hat{z}_m^{(2)} - \lambda_m\hat{z}_m^{(1)}\hat{z}_k^{(1)}\hat{z}_k^{(2)}\right) \quad, \tag{10}$$

*and reach a regime in which the eigenvalues are comparable in magnitude. In this regime, the expected update over augmentations is well approximated by:*

$$\frac{\mathrm{d}\hat{z}_m}{\mathrm{d}t} \approx \eta\lambda_m\cdot\mathbb{E}\left[\frac{\hat{z}_m^2}{\|D\hat{z}\|^3}\right]\cdot\mathbb{E}\left[\frac{\hat{z}_m}{\|\hat{z}\|}\right]\cdot\sum_{k\neq m}\lambda_k\left(\lambda_k - \lambda_m\right), \tag{11}$$

where we have assumed averages over augmentations. See Appendix B for the proof. Theorem 2 states that $\frac{\mathrm{d}}{\mathrm{d}t}\hat{z}_m$ has the same or different sign as $\hat{z}_m$ depending on the sign of the aggregate sum $\sum_{k\neq m}\lambda_k(\lambda_k - \lambda_m)$. This relation suggests that a steady state is only reached through mutual agreement when the non-zero eigenvalues are all equal. In contrast to the Euclidean case, there is no pre-specified target value (see Fig. 5 in Appendix A). Thus, the cosine loss also induces implicit variance regularization, but through a markedly different mechanism in which eigenmodes cooperate.

## 2.3 Stop-grad and predictor network are essential for implicit variance regularization.

We now extend our analysis to explain the known failure modes due to ablating the predictor or the stop-gradient for each distance metric. When we omit the stop-grad operator from $\mathcal{L}^{\text{euc}}$, we have:

$$\mathcal{L}_{\text{noSG}}^{\text{euc}} = \tfrac{1}{2}\|W_{\text{P}}z^{(1)} - z^{(2)}\|^2 \quad\Rightarrow\quad \frac{\mathrm{d}\hat{z}_m}{\mathrm{d}t} = -\eta\left(1 - \lambda_m\right)^2\hat{z}_m \quad, \tag{12}$$

so that $\frac{\mathrm{d}}{\mathrm{d}t}\hat{z}_m$ and $\hat{z}_m$ always have opposite signs (see Appendix C for the derivation). This drives the representations toward zero with exponentially decaying eigenvalues, causing the notorious representational collapse [5]. Omitting the stop-grad operator from $\mathcal{L}^{\text{cos}}$ yields a nontrivial expression for the dynamics causing the largest eigenmode to diverge (see Appendix C). Interestingly, this is different from the collapse to zero inferred for the Euclidean distance.

Similarly, when removing the predictor network in the Euclidean loss case, the dynamics read:

$$\mathcal{L}_{\text{noPred}}^{\text{euc}} = \tfrac{1}{2}\|\boldsymbol{z}^{(1)} - \text{SG}(\boldsymbol{z}^{(2)})\|^2 \quad \Rightarrow \quad \frac{\mathrm{d}\hat{z}_m}{\mathrm{d}t} = 0 \quad, \tag{13}$$

meaning that no learning updates occur. When the predictor is removed in the cosine loss case, the dynamics are:

$$\mathcal{L}_{\text{noPred}}^{\text{cos}} = -\frac{\left(\boldsymbol{z}^{(1)}\right)^\top \text{SG}(\boldsymbol{z}^{(2)})}{\|\boldsymbol{z}^{(1)}\|\|\text{SG}(\boldsymbol{z}^{(2)})\|} \Rightarrow \frac{\mathrm{d}\hat{z}_m}{\mathrm{d}t} = \eta \sum_{k \neq m} \left( \mathbb{E}\left[\frac{\hat{z}_k^2}{\|\hat{\boldsymbol{z}}\|^3}\right] \mathbb{E}\left[\frac{\hat{z}_m}{\|\hat{\boldsymbol{z}}\|}\right] - \mathbb{E}\left[\frac{\hat{z}_m \hat{z}_k}{\|\hat{\boldsymbol{z}}\|^3}\right] \mathbb{E}\left[\frac{\hat{z}_k}{\|\hat{\boldsymbol{z}}\|}\right] \right). \tag{14}$$

As we show in Appendix C, these dynamics also avoid collapse. However, the effective learning rates become impractically small without the eigenvalue factors from Eq. (11). We summarized the predicted dynamics of all settings in Table 1. Thus, our analysis provides mechanistic explanations for why stop-grad and predictor networks are required for avoiding collapse in non-contrastive SSL.

Table 1: Summary of eigendynamics as predicted by our analysis for linear networks.

| Loss | $\mathrm{d}\hat{z}_m/\mathrm{d}t \propto$ | Predicted dynamics |
|---|---|---|
| $\mathcal{L}^{\text{euc}}$ | $\lambda_m(1-\lambda_m)$ | $\lambda$s converge to 1, large ones faster. |
| $\mathcal{L}_{\text{noSG}}^{\text{euc}}$ | $-(1-\lambda_m)^2$ | All $\lambda$s collapse. |
| $\mathcal{L}_{\text{noPred}}^{\text{euc}}$ | $0$ | No learning updates. |
| $\mathcal{L}_{\text{iso}}^{\text{euc}}$ | $(1-\lambda_m)$ | $\lambda$s converge to 1 at homogeneous rates. |
| $\mathcal{L}^{\text{cos}}$ | $\lambda_m \sum_{k \neq m} \lambda_k(\lambda_k - \lambda_m)$ | $\lambda$s converge to equal values. |
| $\mathcal{L}_{\text{noSG}}^{\text{cos}}$ | Appendix C | All $\lambda$s diverge. |
| $\mathcal{L}_{\text{noPred}}^{\text{cos}}$ | Appendix C | $\lambda$s converge to equal values at low rates. |
| $\mathcal{L}_{\text{iso}}^{\text{cos}}$ | $\sum_{k \neq m} \lambda_k(\lambda_k - \lambda_m)$ | $\lambda$s converge to equal values at homogeneous rates. |

## 2.4 Isotropic losses that equalize convergence across eigenmodes

In Eqs. (9) and (11) the eigenvalues appear as multiplicative learning rate modifiers in front of the difference terms that determine the fixed point. Hence, modes with larger eigenvalues converge faster than modes with smaller eigenvalues, reminiscent of previous theoretical work on supervised learning [27]. We hypothesized that the anisotropy in learning dynamics could lead to slow convergence for small eigenvalue modes or instability for large eigenvalues. To alleviate this issue, we designed alternative isotropic loss functions that equalize relaxation dynamics for all eigenmodes by exploiting the stop-grad function. Put simply, this involves taking the dynamics from Eqs. (8) and (10), removing the leading $\lambda_m$ term, and deriving the loss function that would result in the desired dynamics. One such isotropic "IsoLoss" function for the Euclidean distance is:

$$\mathcal{L}_{\text{iso}}^{\text{euc}} = \tfrac{1}{2}\|\boldsymbol{z}^{(1)} - \text{SG}(\boldsymbol{z}^{(2)} + \boldsymbol{z}^{(1)} - W_{\text{P}}\boldsymbol{z}^{(1)})\|^2. \tag{15}$$

We note that this IsoLoss has the same numerical value as $\mathcal{L}^{\text{euc}}$, but the gradient flow is modified by placing the prediction inside the stop-grad and also adding and subtracting $\boldsymbol{z}^{(1)}$ inside and outside of the stop-grad. The associated idealized learning dynamics in our analytic framework are given by:

$$\frac{\mathrm{d}\hat{z}_m}{\mathrm{d}t} = \eta\left(1-\lambda_m\right)\hat{z}_m, \tag{16}$$

where the $\lambda_m$ factor (cf. Eq. (9)) disappeared (Table 1). Similarly, for the cosine distance,

$$\mathcal{L}_{\text{iso}}^{\text{cos}} = -(\boldsymbol{z}^{(1)})^\top \text{SG}\left(\frac{\boldsymbol{z}^{(2)}}{\|W_{\text{P}}\boldsymbol{z}^{(1)}\|\|\boldsymbol{z}^{(2)}\|}\right) + \frac{1}{2}\text{SG}\left(\frac{(W_{\text{P}}\boldsymbol{z}^{(1)})^\top \boldsymbol{z}^{(2)}}{\|W_{\text{P}}\boldsymbol{z}^{(1)}\|^3\|\boldsymbol{z}^{(2)}\|}\right)\|W_{\text{P}}^{1/2}\boldsymbol{z}^{(1)}\|^2 \tag{17}$$

is one possible IsoLoss, in which $W_{\text{P}}^{1/2} = U D^{1/2} U^\top$ with the square-root applied element-wise to the diagonal matrix $D$. While this IsoLoss does not preserve numerical equality with the original loss $\mathcal{L}^{\text{cos}}$, it achieves the desired effect of removing the leading $\lambda_m$ learning-rate modifier (cf. Table 1).

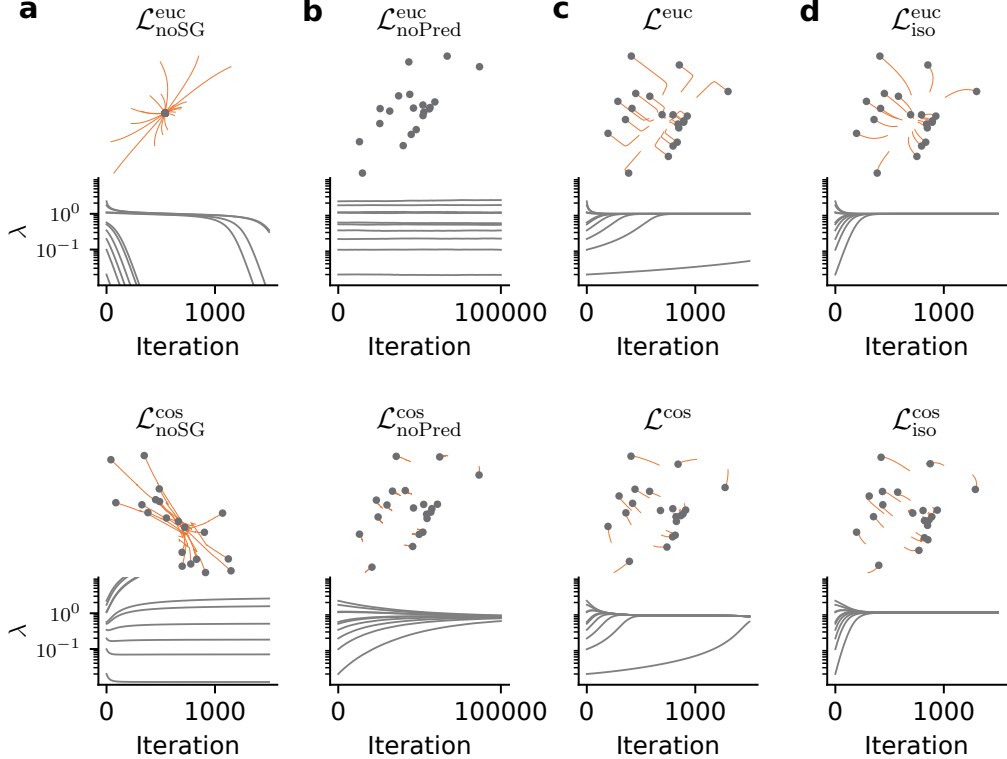

Figure 2: Evolution of representations (top) and eigenvalues (below) of $W_{\mathrm{P}}$ throughout training with different loss functions. The representational trajectories correspond to training with $M = 2$ for visualization and the points signify the final network outputs. The eigenvalues were computed with dimensions $N = 15$ and $M = 10$. **(a)** Omitting the stop-grad leads to representational collapse in the Euclidean case (top), and diverging eigenvalues for the cosine case (bottom). **(b)** No learning occurs without the predictor with the Euclidean distance, but learning does occur with the cosine distance, although at low rates. Note the change in scale of the time-axis. **(c)** Optimizing the BYOL/SimSiam loss leads to isotropic representations under both distance metrics. **(d)** Optimizing IsoLoss has the same effect, but with uniform convergence dynamics for all eigenvalues for both distance metrics.

## 3   Numerical experiments

To validate our theoretical findings (cf. Table 1), we first simulated a small linear Siamese neural network as shown in Fig.1a, for which Theorems 1 and 2 hold exactly. We fed the network with independent standard Gaussian inputs, and generated pairs of augmentations using isotropic Gaussian perturbations of standard deviation $\sigma = 0.1$. We then trained the linear encoder with each configuration described above. Training the network with $\mathcal{L}_{\mathrm{noSG}}^{\mathrm{euc}}$ resulted in collapse with exponentially decaying eigenvalues, whereas $\mathcal{L}_{\mathrm{noSG}}^{\mathrm{cos}}$ succumbed to diverging eigenvalues as predicted (Fig. 2a). Training without the predictor caused vanishing updates for $\mathcal{L}_{\mathrm{noPred}}^{\mathrm{euc}}$ and slow learning for $\mathcal{L}_{\mathrm{noPred}}^{\mathrm{cos}}$, in line with our analysis (Fig. 2b). Optimizing $\mathcal{L}^{\mathrm{euc}}$, the representations become increasingly isotropic with all the eigenvalues $\lambda_m$ converging to one (Fig. 2c, top), whereas optimizing $\mathcal{L}^{\mathrm{cos}}$ also resulted in the eigenvalues converging to the same value, but different from one (Fig. 2c, bottom). The anisotropy in the dynamics of different eigenmodes noted above is particularly striking in the case of the Euclidean distance (Fig. 2c). Training with $\mathcal{L}_{\mathrm{iso}}^{\mathrm{euc}}$ and $\mathcal{L}_{\mathrm{iso}}^{\mathrm{cos}}$ resulted in similar convergence properties as their non-isotropic counterparts, but the eigenmodes converged at more homogeneous rates (Fig. 2d). Finally, we confirmed that these findings were qualitatively similar in the corresponding nonlinear networks with ReLU activation (see Fig. 6 in Appendix A). Thus, our theoretical findings hold up in simple Siamese networks.

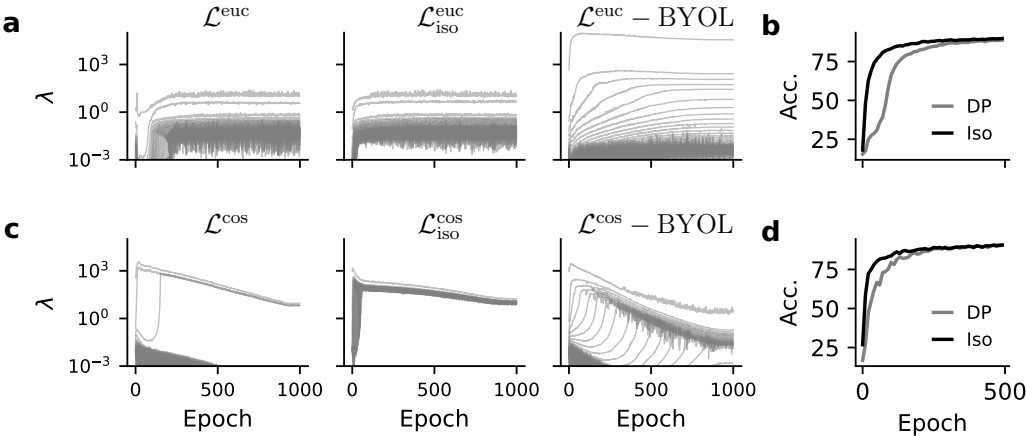

Figure 3: Learning dynamics for a ResNet-18 network trained with different loss functions. **(a)** Evolution of the eigenvalues of the representation correlation matrix during training for closed-form predictors as prescribed by DirectPred (left) and IsoLoss (center). Right: Standard BYOL with the nonlinear trainable predictor. For clarity, we plot only one in ten eigenvalues. Both $\mathcal{L}^{euc}$ and $\mathcal{L}^{euc}_{iso}$ drive the eigenvalues to converge quickly and remain constant thereafter with relatively small fluctuations (note the logarithmic scale). BYOL results in the eigenvalues being spread across a large range of magnitudes. **(b)** Linear readout validation accuracy for $\mathcal{L}^{euc}$ and $\mathcal{L}^{euc}_{iso}$ during the first 500 training epochs. IsoLoss accelerates the initial learning dynamics as predicted by the theory. **(c)** Same as in (a) but for the cosine distance. $\mathcal{L}^{cos}$ recruits few large eigenvalues, but drives them gradually to the same magnitude, whereas $\mathcal{L}^{cos}_{iso}$ quickly recruits *all* eigenvalues and causing them to converge to an isotropic solution. In contrast, BYOL recruits eigenvalues in a step-wise manner. **(d)** Same as (b) but for the cosine distance.

## 3.1 Theory qualitatively captures dynamics in nonlinear networks and real-world datasets.

To investigate how well our theoretical analysis holds up in non-toy settings, we performed several self-supervised learning experiments on CIFAR-10, CIFAR-100 [29], STL-10 [30], and TinyImageNet [31]. We based our implementation[1] on the Solo-learn library [32], and used a ResNet-18 backbone [33] as the encoder and the cosine loss, unless mentioned otherwise (see Appendix D for details). As baselines for comparison, we trained the same backbone using BYOL with the nonlinear predictor and DirectPred with the closed-form linear predictor. We recorded the online readout accuracy of a linear classifier trained on frozen features following standard practice, evaluated either on the held-out validation or test set where available.

We found that the eigenvalue dynamics of the representational correlation matrix in the ResNet-18 closely mirrored the analytical predictions for the closed-form predictor. For Euclidean distances (Fig. 3a), the eigenvalues for DirectPred and IsoLoss converged to a small range of values around one. However, the dynamics for BYOL with a learnable nonlinear predictor deviated significantly with the eigenvalues distributed over a larger range. Consistent with our analysis, IsoLoss had faster initial dynamics for the eigenvalues which also resulted in a faster initial improvements in model performance (Fig. 3b). The faster learning with IsoLoss was even more evident for the cosine distance (Fig. 3c). Surprisingly, BYOL, which uses a nonlinear predictor also closely matched the predicted dynamics in the case of the cosine distance. Furthermore, the dynamics showed a stepwise learning phenomenon wherein eigenvalues are progressively recruited one-by-one, consistent with recent findings for other SSL methods [34]. Finally, IsoLoss exhibited faster initial learning (Fig. 3d), in agreement with our theoretical analysis. Thus, our theoretical analysis accurately predicts key properties of the eigenvalue dynamics in nonlinear networks trained on real-world datasets.

---

[1]Code is available at `https://github.com/fmi-basel/implicit-var-reg`

## 3.2 IsoLoss promotes eigenvalue recruitment and works without an EMA target network.

To further investigate the impact of IsoLoss on learning, we first verified that it does not have any adverse effects on downstream classification performance. We found that IsoLoss matched or outperformed DirectPred on all benchmarks (Table 2) when trained with an EMA target network as used in the original studies. Yet, it performed slightly worse than BYOL, which uses a nonlinear predictor and an EMA target network. Because EMA target networks are thought to amplify small eigenvalues [17], we speculated that IsoLoss may work without it. We repeated training for the closed-form predictor losses without EMA to test this idea. We found that $\mathcal{L}_{\mathrm{iso}}^{\mathrm{cos}}$ was indeed robust to EMA removal. However, it caused a slight drop in performance (Table 2) and a notable reduction in the recruitment of small eigenvalues (see Fig. 7 in Appendix A). In contrast, optimizing the standard BYOL/SimSiam loss $\mathcal{L}^{\mathrm{cos}}$ with the symmetric linear predictor was unstable, as reported previously [17]. Finally, we confirmed the above findings also hold for $\alpha = 1$ (cf. Eq. (1)) as prescribed by DirectCopy [18] (see Table 3 in Appendix A). Thus, IsoLoss allows training without an EMA target network.

Table 2: Linear readout validation accuracy in % $\pm$ stddev over five random seeds. The $\dagger$ denotes crashed runs, known to occur with symmetric predictors like DirectPred [17]. Starred values $^*$ were taken from the Solo-learn library [32].

| Model | EMA | CIFAR-10 | CIFAR-100 | STL-10 | TinyImageNet |
|---|---|---|---|---|---|
| BYOL | Yes | $92.6^*$ | $70.5^*$ | $91.7 \pm 0.1$ | $38.3 \pm 1.5$ |
| SimSiam | No | $90.7 \pm 0.2$ | $66.3 \pm 0.4$ | $87.5 \pm 0.7$ | $39.8 \pm 0.6$ |
| DirectPred ($\alpha = 0.5$) | Yes | $92.0 \pm 0.2$ | $66.6 \pm 0.5$ | $88.8 \pm 0.3$ | $40.1 \pm 0.5$ |
| | No | $12.1 \pm 1.3^\dagger$ | $1.6 \pm 0.6^\dagger$ | $10.4 \pm 0.1^\dagger$ | $1.3 \pm 0.2^\dagger$ |
| IsoLoss (ours) | Yes | $91.5 \pm 0.2$ | $69.0 \pm 0.2$ | $89.0 \pm 0.3$ | $44.8 \pm 0.4$ |
| | No | $91.5 \pm 0.2$ | $64.3 \pm 0.3$ | $87.4 \pm 0.1$ | $40.4 \pm 0.4$ |

The above result suggests that IsoLoss promotes the recruitment of small eigenvalues in closed-form predictors. Another factor that has been implicated in suppressing recruitment is weight decay [18]. To probe how weight decay and IsoLoss affect small eigenvalue recruitment, we repeated the above simulations with EMA and different amounts of weight decay. Indeed, we observed less eigenvalue recruitment with increasing weight decay for DirectPred (Appendix A, Fig. 8a), but not for IsoLoss (Fig. 8b). However, for IsoLoss larger weight decay resulted in lower magnitudes of *all* eigenvalues. Hence, IsoLoss reduces the impact of weight decay on eigenvalue recruitment.

## 4 Discussion

We provided a comprehensive analysis of the SSL representational dynamics in the eigenspace of closed-form linear predictor networks (i.e., DirectPred and DirectCopy) for both the Euclidean loss and the more commonly used cosine similarity. Our analysis revealed how asymmetric losses prevent representational and dimensional collapse through *implicit* variance regularization along orthogonal eigenmodes, thereby formally linking predictor-based SSL with explicit variance regularization approaches [6, 7, 14]. Our work provides a theory framework which further complements the growing body of work linking contrastive and non-contrastive SSL [24, 35–38].

We empirically validated the key predictions of our analysis in linear and nonlinear network models on several datasets, including CIFAR-10/100, STL-10, and TinyImageNet. Moreover, we found that the eigenvalues of the predictor network act as learning rate multipliers, causing anisotropic learning dynamics. We derived Euclidean and cosine IsoLosses, which counteract this anisotropy and enable closed-form linear predictor methods to work without an EMA target network, thereby further consolidating its presumed role in boosting small eigenvalues [17].

To our knowledge, this is the first work to comprehensively characterize asymmetric SSL learning dynamics for the cosine distance metric widely used in practice. However, our analysis rests on several assumptions. First, the analytic link through the NTK between gradient descent on parameters and the representational changes is an approximation in nonlinear networks. Moreover, we assumed Gaussian i.i.d inputs for proving Theorems 1 and 2. Although these assumptions generally do not

hold in nonlinear networks, our analysis qualitatively captures their overall learning behavior and predicts how networks respond to changes in the stop-grad placement.

In summary, we have provided a simple theoretical explanation of how asymmetric loss configurations prevent representational collapse in SSL and elucidate their inherent dependence on the placement of the stop-grad operation. We further demonstrated how the eigenspace framework allows crafting new loss functions with a distinct impact on the SSL learning dynamics. We provided one specific example of such loss functions, IsoLoss, which equalizes the learning dynamics in the predictor's eigenspace, resulting in faster initial learning and improved stability. In contrast to DirectPred, IsoLoss learns stably without an EMA target network. Our work thus lays out an effective framework for analyzing and developing new SSL loss functions.

## Acknowledgments and Disclosure of Funding

This project was supported by the Swiss National Science Foundation [grant numbers PCEFP3_202981 and TMPFP3_210282], by EU's Horizon Europe Research and Innovation Programme (grant agreement number 101070374) funded through SERI (ref 1131-52302), and the Novartis Research Foundation. The authors declare no competing interests.

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

# A   Supplementary Material

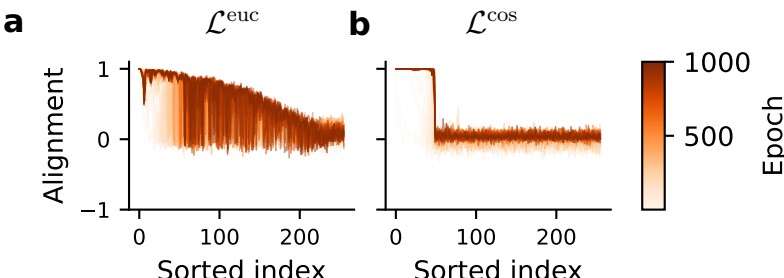

Figure 4:  Eigenspace alignment between $C_{\boldsymbol{z}} = \mathbb{E}_{\boldsymbol{x}}\left[\boldsymbol{z}\boldsymbol{z}^{\top}\right]$ and a linear predictor $W_{\mathrm{P}}$ trained with gradient descent. Following [1], we measure eigenspace alignment as the cosine between $\boldsymbol{u}_i$ and $W_{\mathrm{P}}\boldsymbol{u}_i$ for every eigenvector $\boldsymbol{u}_i$ of $C_{\boldsymbol{z}}$. **(a)** Measured alignment for every eigenvector of $C_{\boldsymbol{z}}$ ordered by sorted eigenvalue indices over training epochs, when the network is trained with $\mathcal{L}^{\mathrm{euc}}$. **(b)** Same as **(a)** but for training with $\mathcal{L}^{\mathrm{cos}}$.

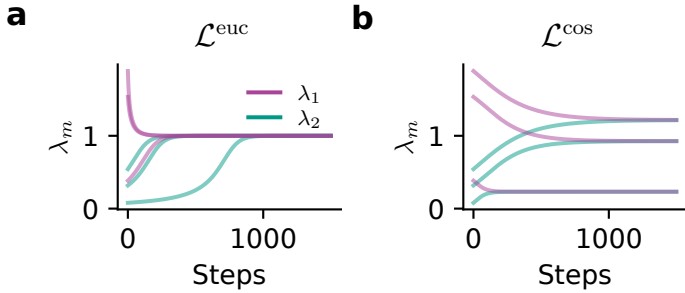

Figure 5:  Comparison between the dynamics under Euclidean and cosine asymmetric losses for different initializations in a network with $M = 2$ output neurons. **(a)** Observed dynamics of the eigenvalues in the two-neuron toy network under three different initializations. Both eigenvalues always converge to 1 regardless of the initialization. **(b)** Same as **(a)**, but for the cosine distance. Under different initializations, the two eigenvalues converge to arbitrary, but equal, values.

Table 3: Linear readout validation accuracy in % $\pm$ stddev over five random seeds. The $\dagger$ denotes crashed runs, known to occur with symmetric predictors like DirectPred [1].

| Model | $\alpha$ | EMA | CIFAR-10 | CIFAR-100 | STL-10 | TinyImageNet |
|---|---|---|---|---|---|---|
| DirectCopy | 1 | Yes | $91.3 \pm 0.2$ | $68.7 \pm 0.3$ | $89.3 \pm 0.2$ | $45.3 \pm 0.3$ |
| | | No | $12.1 \pm 0.6^{\dagger}$ | $1.5 \pm 0.5^{\dagger}$ | $10.3 \pm 0.1^{\dagger}$ | $0.6 \pm 0.1^{\dagger}$ |
| DirectPred | 0.5 | Yes | $92.0 \pm 0.2$ | $66.6 \pm 0.5$ | $88.8 \pm 0.3$ | $40.1 \pm 0.5$ |
| | | No | $12.1 \pm 1.3^{\dagger}$ | $1.6 \pm 0.6^{\dagger}$ | $10.4 \pm 0.1^{\dagger}$ | $1.3 \pm 0.2^{\dagger}$ |
| IsoLoss (ours) | 1 | Yes | $91.5 \pm 0.2$ | $69.3 \pm 0.2$ | $89.1 \pm 0.3$ | $45.6 \pm 0.9$ |
| | | No | $91.4 \pm 0.2$ | $63.0 \pm 0.3$ | $86.9 \pm 0.4$ | $38.5 \pm 0.5$ |
| | 0.5 | Yes | $91.5 \pm 0.2$ | $69.0 \pm 0.2$ | $89.0 \pm 0.3$ | $44.8 \pm 0.4$ |
| | | No | $91.5 \pm 0.2$ | $64.3 \pm 0.3$ | $87.4 \pm 0.1$ | $40.4 \pm 0.4$ |

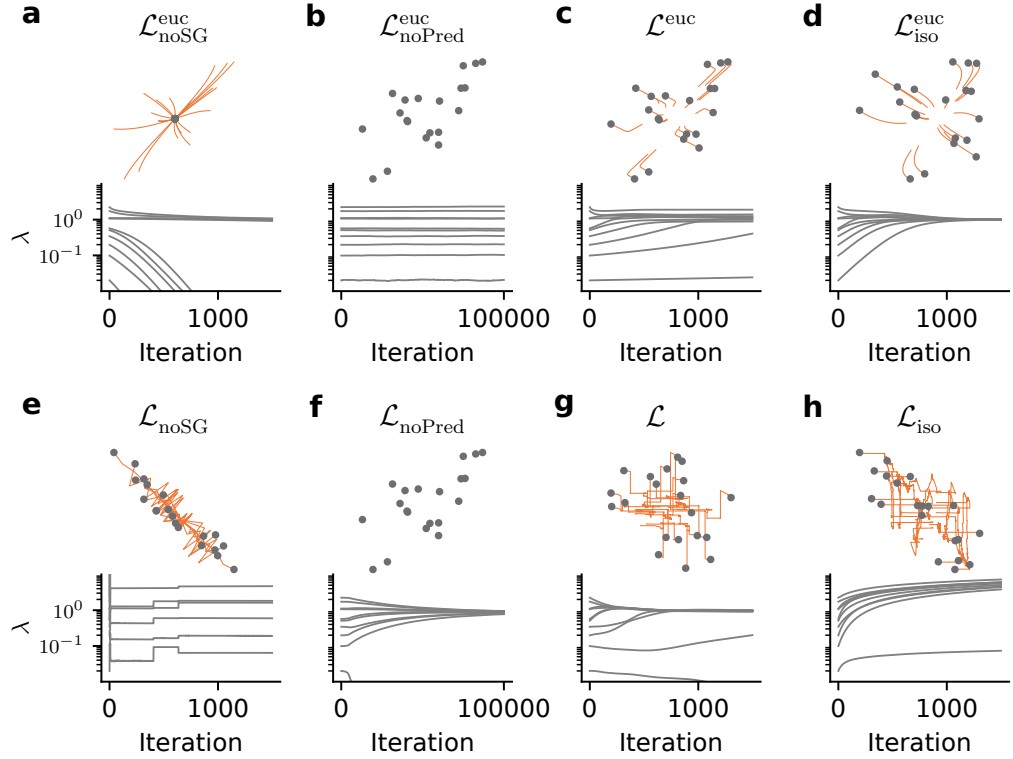

Figure 6: Same as Fig. 2 but with a ReLU nonlinearity on the embeddings. We observe learning dynamics qualitatively similar to the linear network.

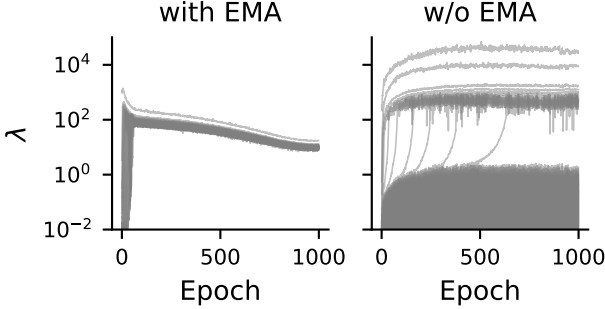

Figure 7: Eigenvalue dynamics for learning under IsoLoss ($\mathcal{L}_{\mathrm{iso}}^{\mathrm{cos}}$) with and without the EMA target network. Removing the EMA results in markedly different dynamics with fewer eigenmodes recruited during training.

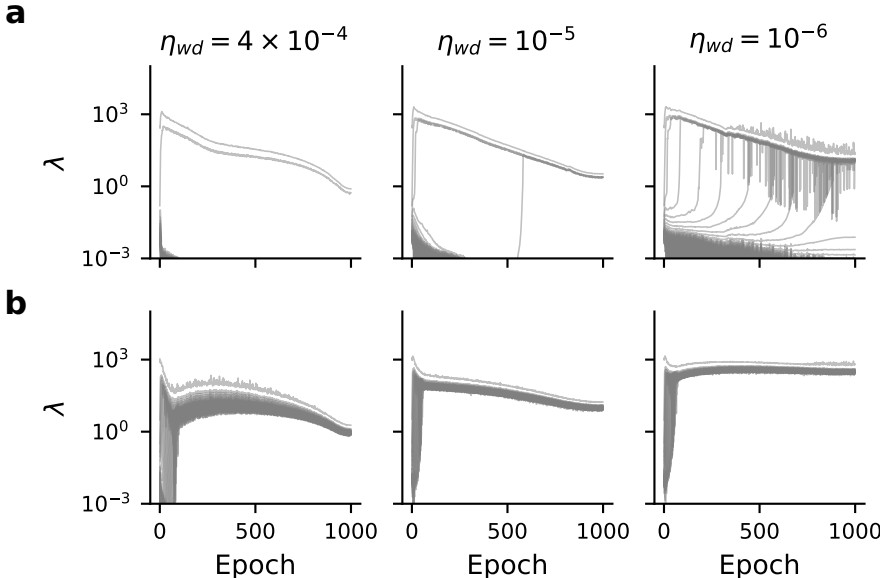

Figure 8: Effect of weight decay on eigenvalue recruitment for DirectPred and IsoLoss. **(a)** Evolution of eigenvalues during learning under DirectPred ($\mathcal{L}^{\cos}$) with EMA and different amounts of weight decay. Decreasing weight decay correlates with the number of eigenvalues recruited during learning. **(b)** Same as **(a)** but for IsoLoss ($\mathcal{L}^{\cos}_{\mathrm{iso}}$). All eigenvalues are recruited independent of the strength of weight decay. However, the magnitude of the eigenvalues inversely correlates with the magnitude of weight-decay.

## B  Proofs

**Lemma 1.** (Euclidean and cosine loss in the predictor eigenspace) *Let $W_\mathrm{P}$ be a linear predictor set according to DirectPred with eigenvalues $\lambda_m$, and $\hat{z}$ the representations expressed in the predictor's eigenbasis. Then the asymmetric Euclidean loss $\mathcal{L}^\mathrm{euc}$ and cosine loss $\mathcal{L}^\mathrm{cos}$ can be expressed as:*

$$\mathcal{L}^\mathrm{euc} = \tfrac{1}{2} \sum_m^M |\lambda_m \hat{z}_m^{(1)} - \mathrm{SG}(\hat{z}_m^{(2)})|^2 \quad , \tag{3}$$

$$\mathcal{L}^\mathrm{cos} = -\sum_m^M \frac{\lambda_m \hat{z}_m^{(1)} \mathrm{SG}(\hat{z}_m^{(2)})}{\|D\hat{\boldsymbol{z}}^{(1)}\|\|\mathrm{SG}(\hat{\boldsymbol{z}}^{(2)})\|} \quad . \tag{4}$$

*Proof.* Under DirectPred, the predictor is a symmetric matrix with eigendecomposition $W_\mathrm{P} = UDU^\top$. Since $U$ is an orthogonal matrix, we also have $UU^\top = I$ so that we can simplify the losses as follows:

$$\begin{aligned}
\mathcal{L}^\mathrm{euc} &= \tfrac{1}{2}\|W_\mathrm{P}\boldsymbol{z}^{(1)} - \mathrm{SG}(\boldsymbol{z}^{(2)})\|^2 \\
&= \tfrac{1}{2}\|UDU^\top \boldsymbol{z}^{(1)} - \mathrm{SG}(UU^\top \boldsymbol{z}^{(2)})\|^2 \\
&= \tfrac{1}{2}\|D\hat{\boldsymbol{z}}^{(1)} - \mathrm{SG}(\hat{\boldsymbol{z}}^{(2)})\|^2 \\
&= \tfrac{1}{2}\sum_m^M |\lambda_m \hat{z}_m^{(1)} - \mathrm{SG}(\hat{z}_m^{(2)})|^2
\end{aligned}$$

$$\begin{aligned}
\mathcal{L} &= -\frac{\left(W_\mathrm{P}\boldsymbol{z}^{(1)}\right)^\top \mathrm{SG}(\boldsymbol{z}^{(2)})}{\|W_\mathrm{P}\boldsymbol{z}^{(1)}\|\|\mathrm{SG}(\boldsymbol{z}^{(2)})\|} \\
&= -\frac{(\boldsymbol{z}^{(1)})^\top UDU^\top \mathrm{SG}(\boldsymbol{z}^{(2)})}{\|UDU^\top \boldsymbol{z}^{(1)}\|\|\mathrm{SG}(UU^\top \boldsymbol{z}^{(2)})\|} \\
&= -\frac{(\hat{\boldsymbol{z}}^{(1)})^\top D\, \mathrm{SG}(\hat{\boldsymbol{z}}^{(2)})}{\|D\hat{\boldsymbol{z}}^{(1)}\|\|\mathrm{SG}(\hat{\boldsymbol{z}}^{(2)})\|} \\
&= -\sum_m^M \frac{\lambda_m \hat{z}_m^{(1)}\, \mathrm{SG}(\hat{z}_m^{(2)})}{\|D\hat{\boldsymbol{z}}^{(1)}\|\|\mathrm{SG}(\hat{\boldsymbol{z}}^{(2)})\|} \quad ,
\end{aligned}$$

where we used the fact that $U$ is orthogonal and therefore does not change the Euclidean norm. $\hat{\boldsymbol{z}} = U^\top \boldsymbol{z}$ is the representation rotated into the eigenbasis. $\qquad\square$

**Lemma 2.** (Learning dynamics in a rotated basis) *Assuming that a given loss $\mathcal{L}$ is optimized by gradient descent on the parameters of a neural network with network outputs $\boldsymbol{z}$, a given orthogonal transformation $\hat{\boldsymbol{z}} = U^\top z$ and learning rate $\eta$, then the rotated representations $\hat{\boldsymbol{z}}$ evolve according to the dynamics:*

$$\frac{\mathrm{d}\hat{\boldsymbol{z}}}{\mathrm{d}t} = -\eta \hat{\Theta}_t(\boldsymbol{x}, \mathcal{X}) \nabla_{\hat{\boldsymbol{z}}} \mathcal{L} \quad ,$$

*where $\hat{\Theta}_t(\mathcal{X}, \mathcal{X}) = \nabla_{\boldsymbol{\theta}} \hat{\mathcal{Z}} \nabla_{\boldsymbol{\theta}} \hat{\mathcal{Z}}^\top$ is the empirical NTK expressed in the rotated basis.*

*Proof.* Let $\boldsymbol{\theta}$ be the parameters of the neural network. Then we obtain the representational dynamics using the chain rule in the continuous-time gradient-flow setting [2]:

$$\begin{aligned}
\frac{\mathrm{d}\hat{\boldsymbol{z}}}{\mathrm{d}t} &= \nabla_{\boldsymbol{\theta}} \hat{\boldsymbol{z}} \frac{\mathrm{d}\boldsymbol{\theta}}{\mathrm{d}t} \\
&= \nabla_{\boldsymbol{\theta}} \hat{\boldsymbol{z}} \left( -\eta \nabla_{\boldsymbol{\theta}} \mathcal{L} \right) \\
&= \nabla_{\boldsymbol{\theta}} \hat{\boldsymbol{z}} \left( -\eta \nabla_{\boldsymbol{\theta}} \hat{\mathcal{Z}}^\top \nabla_{\hat{\boldsymbol{z}}} \mathcal{L} \right) \\
&= -\eta \hat{\Theta}_t(x, \mathcal{X}) \nabla_{\hat{\boldsymbol{z}}} \mathcal{L} \quad .
\end{aligned}$$

The above is a reiteration of the derivation of Eq. (2) given by Lee et al. [2], with an additional orthogonal transformation on the network outputs. $\square$

We proceed by proving the following Lemma which we will use in our proofs of Theorems 1 and 2.

**Lemma 3.** *The NTK for a linear network is invariant under orthogonal transformations of the network output.*

*Proof.* We first note that for a linear network, the parameters $\boldsymbol{\theta}$ are just the feedforward weights $W$. Therefore, for any orthogonal transformation $U$ of the network output:

$$\begin{aligned}
\hat{\boldsymbol{z}} = U^\top f(\boldsymbol{x}) &= U^\top W \boldsymbol{x} \\
\Rightarrow \nabla_{\boldsymbol{\theta}} \hat{\boldsymbol{z}} = \nabla_W \hat{\boldsymbol{z}} = \nabla_W \left( U^\top W \boldsymbol{x} \right) &= \boldsymbol{x}^\top \otimes U^\top,
\end{aligned} \tag{18}$$

where $\otimes$ is the Kronecker product resulting from the fact that every input vector component appears in the update once for each output component.

We now study $\hat{\Theta}_t(\mathcal{X}, \mathcal{X})$, the transformed empirical NTK (cf. Lemma 2). The $(M \times M)$ diagonal blocks in the full $(M|\mathsf{D}| \times M|\mathsf{D}|)$ empirical NTK $\hat{\Theta}_t(\mathcal{X}, \mathcal{X})$ correspond to single samples and the off-diagonal blocks are cross-terms between samples, where $|\mathsf{D}|$ denotes the size of the training dataset and $M$ the dimension of the outputs. We can develop a generic expression for each $(M \times M)$ block $\hat{\Theta}_t(\boldsymbol{x}_i, \boldsymbol{x}_j)$ corresponding to the interactions between samples $i$ and $j$ as:

$$\begin{aligned}
\hat{\Theta}_t(\boldsymbol{x}_i, \boldsymbol{x}_j) &= \nabla_W \hat{\boldsymbol{z}}_i \nabla_W \hat{\boldsymbol{z}}_j^\top \\
&= \left( \boldsymbol{x}_i^\top \otimes U^\top \right) \left( \boldsymbol{x}_j^\top \otimes U^\top \right)^\top \\
&= \left( \boldsymbol{x}_i^\top \otimes U^\top \right) \left( \boldsymbol{x}_j \otimes U \right) \\
&= \left( \boldsymbol{x}_i^\top \boldsymbol{x}_j \right) \otimes \left( U^\top U \right) \\
&= \left( \boldsymbol{x}_i^\top \boldsymbol{x}_j \right) \otimes I_{\mathsf{M}} \\
&= \left( \boldsymbol{x}_i^\top \boldsymbol{x}_j \right) I_{\mathsf{M}}.
\end{aligned} \tag{19}$$

where we have used the fact that $(A \otimes B)^\top = A^\top \otimes B^\top$ and $(A \otimes B)(C \otimes D) = AC \otimes BD$. Here, $I_{\mathsf{M}}$ is the identity matrix of size $M$. Noting that Eq. (19) is unchanged when $U$ is just the identity matrix completes the proof. $\square$

**Theorem 1.** (Representational dynamics under $\mathcal{L}^{\text{euc}}$) *For a linear network with i.i.d Gaussian inputs learning with $\mathcal{L}^{\text{euc}}$, the representational dynamics of each mode $m$ independently follow the gradient of the loss $-\nabla_{\hat{z}}\mathcal{L}^{\text{euc}}$. More specifically, the dynamics uncouple and follow $M$ independent differential equations:*

$$\frac{\mathrm{d}\hat{z}_m^{(1)}}{\mathrm{d}t} = -\eta \frac{\partial \mathcal{L}^{\text{euc}}}{\partial \hat{z}_m^{(1)}}(t) = \eta \lambda_m \left( \hat{z}_m^{(2)} - \lambda_m \hat{z}_m^{(1)} \right) \quad , \tag{8}$$

*which, after taking the expectation over augmentations yields the dynamics:*

$$\frac{\mathrm{d}\hat{z}_m}{\mathrm{d}t} = \eta \lambda_m \left(1 - \lambda_m\right) \hat{z}_m \quad . \tag{9}$$

*Proof.* For a linear network with weights $W \in \mathbb{R}^{M \times N}$, we have from Lemma 3 that the empirical NTK $\hat{\Theta}(\mathcal{X}, \mathcal{X})$ in the orthogonal eigenbasis is equal to the empirical NTK $\Theta(\mathcal{X}, \mathcal{X})$ in the original basis. Furthermore from the proof for the lemma (see Eq. (19) above), each $(M \times M)$ block of the full $(M|\mathtt{D}| \times M|\mathtt{D}|)$ empirical NTK is given by:

$$\hat{\Theta}_t(\boldsymbol{x}_i, \boldsymbol{x}_j) = \left(\boldsymbol{x}_i^\top \boldsymbol{x}_j\right) I_{\mathrm{M}}. \tag{20}$$

where $I_M \in \mathbb{R}^{M \times M}$ is the identity. Eq. (20) gives the total effective interaction between the samples $i$ and $j$ from the dataset. For high-dimensional inputs $\boldsymbol{x}$ drawn from an i.i.d standard Gaussian distribution, we have $\boldsymbol{x}_i^\top \boldsymbol{x}_j \approx \delta_{ij}$ by the central limit theorem. Therefore, in the special case of a linear network with Gaussian i.i.d inputs, the representational dynamics (Lemma 2) simplify as follows:

$$\begin{aligned}
\frac{\mathrm{d}\hat{\boldsymbol{z}}_i^{(1)}}{\mathrm{d}t} &= -\eta \hat{\Theta}_t(\boldsymbol{x}_i, \mathcal{X}) \nabla_{\hat{\mathcal{Z}}} \mathcal{L} \\
&= -\eta \hat{\Theta}_t(\boldsymbol{x}_i, \boldsymbol{x}_i) \nabla_{\hat{\boldsymbol{z}}_i} \mathcal{L} - \eta \sum_{j \neq i} \hat{\Theta}_t(\boldsymbol{x}_i, \boldsymbol{x}_j) \nabla_{\hat{\boldsymbol{z}}_j} \mathcal{L} \\
&= -\eta \left( \boldsymbol{x}_i^\top \boldsymbol{x}_i \right) \nabla_{\hat{\boldsymbol{z}}_i} \mathcal{L} - \eta \sum_{j \neq i} \left( \boldsymbol{x}_i^\top \boldsymbol{x}_j \right) \nabla_{\hat{\boldsymbol{z}}_j} \mathcal{L} \\
&= -\eta \nabla_{\hat{\boldsymbol{z}}_i} \mathcal{L} \quad .
\end{aligned} \tag{21}$$

While the assumption of Gaussian i.i.d inputs is quite restrictive, we offer a generalizing interpretation here. Specifically, the above argument also holds when the inputs $\boldsymbol{x}$ are not all mutually orthogonal, but fall into $P$ orthogonal clusters in the input dataset. Then, we would have $\boldsymbol{x}_i^\top \boldsymbol{x}_j \approx \delta_{p_i = p_j}$ where $p_i$ is the "label" of the cluster corresponding to sample $i$. If $\mathcal{P}_i$ is the number of all the samples with the same label $p_i$, then Eq. (21) would simply be scaled to give $\frac{\mathrm{d}\hat{\boldsymbol{z}}_i^{(1)}}{\mathrm{d}t} = -\eta \mathcal{P}_i \nabla_{\hat{\boldsymbol{z}}_i} \mathcal{L}$.

For brevity, we proceed with the simplest case Eq. (21) in which every input is orthogonal. For $\mathcal{L}^{\text{euc}}$, the representational gradient $\nabla_{\hat{\boldsymbol{z}}_i} \mathcal{L}$ is then given by:

$$\nabla_{\hat{\boldsymbol{z}}_i} \mathcal{L}^{\text{euc}} = \left( D_t \hat{\boldsymbol{z}}_i^{(1)} - \hat{\boldsymbol{z}}_i^{(2)} \right) D_t$$

Noting that $D_t$ is just a diagonal matrix containing the eigenvalues $\lambda_m$ and dropping the sample subscript $i$ for notational ease, we obtain for the $m$-th component of $\nabla_{\hat{\boldsymbol{z}}_i} \mathcal{L}^{\text{euc}}$:

$$\frac{\partial \mathcal{L}^{\text{euc}}}{\partial \hat{z}_m} = \lambda_m (\lambda_m \hat{z}_m^{(1)} - \hat{z}_m^{(2)}) \quad .$$

Substituting this result in Eq. (21) gives us Eq. (8), the expression we were looking for. Finally, introducing $\hat{z}_m \equiv \mathbb{E}[\hat{z}_m^{(1)}] = \mathbb{E}[\hat{z}_m^{(2)}]$ as the expectation over augmentations, we find that each eigenmode evolves independently in expectation value as:

$$\begin{aligned}
\mathbb{E}\left[ \frac{\mathrm{d}\hat{z}_m^{(1)}}{\mathrm{d}t} \right] = \frac{\mathrm{d}\hat{z}_m}{\mathrm{d}t} &= \eta \lambda_m \left( \mathbb{E}[\hat{z}_m^{(2)}] - \lambda_m \mathbb{E}[\hat{z}_m^{(1)}] \right) \\
&= \eta \lambda_m \left(1 - \lambda_m\right) \hat{z}_m \quad .
\end{aligned}$$

$\square$

**Theorem 2.** (Representational dynamics under $\mathcal{L}^{\cos}$) *For a linear network with i.i.d Gaussian inputs trained with $\mathcal{L}^{\cos}$, the dynamics follow a system of $M$ coupled differential equations:*

$$\frac{\mathrm{d}\hat{z}_m^{(1)}}{\mathrm{d}t} = \eta \frac{\lambda_m}{\|D\hat{z}^{(1)}\|^3 \|\hat{z}^{(2)}\|} \sum_{k\neq m} \lambda_k \left( \lambda_k (\hat{z}_k^{(1)})^2 \hat{z}_m^{(2)} - \lambda_m \hat{z}_m^{(1)} \hat{z}_k^{(1)} \hat{z}_k^{(2)} \right) \quad , \tag{10}$$

*and reach a regime in which the eigenvalues are comparable in magnitude. In this regime, the expected update over augmentations is well approximated by:*

$$\frac{\mathrm{d}\hat{z}_m}{\mathrm{d}t} \approx \eta \lambda_m \cdot \mathbb{E}\left[ \frac{\hat{z}_m^2}{\|D\hat{z}\|^3} \right] \cdot \mathbb{E}\left[ \frac{\hat{z}_m}{\|\hat{z}\|} \right] \cdot \sum_{k\neq m} \lambda_k (\lambda_k - \lambda_m), \tag{11}$$

*Proof.* We can retrace the steps from the proof for Theorem 1 until Eq. (21):

$$\frac{\mathrm{d}\hat{\boldsymbol{z}}_i^{(1)}}{\mathrm{d}t} = -\eta \nabla_{\hat{\boldsymbol{z}}_i} \mathcal{L} \quad .$$

$\nabla_{\hat{\boldsymbol{z}}_i} \mathcal{L}$ is a vector of dimension $M$. Ignoring the sample subscript $i$ for simplicity, and focusing on the $m$-th component of $\nabla_{\hat{\boldsymbol{z}}_i} \mathcal{L}$, we get:

$$\mathcal{L}^{\cos} = -\sum_m^M \frac{\lambda_m \hat{z}_m^{(1)} \mathrm{SG}(\hat{z}_m^{(2)})}{\|D\hat{\boldsymbol{z}}^{(1)}\| \|\mathrm{SG}(\hat{\boldsymbol{z}}^{(2)})\|}$$

$$\Rightarrow \frac{\partial \mathcal{L}}{\partial \hat{z}_m^{(1)}} = -\frac{\lambda_m \hat{z}_m^{(2)}}{\|D\hat{\boldsymbol{z}}^{(1)}\| \|\hat{\boldsymbol{z}}^{(2)}\|} + \frac{\sum_k \lambda_k \hat{z}_k^{(1)} \hat{z}_k^{(2)}}{\|D\hat{\boldsymbol{z}}^{(1)}\|^3 \|\hat{\boldsymbol{z}}^{(2)}\|} \cdot \lambda_m^2 \hat{z}_m^{(1)}$$

$$= -\frac{\lambda_m}{\|D\hat{\boldsymbol{z}}^{(1)}\|^3 \|\hat{\boldsymbol{z}}^{(2)}\|} \left[ \|D\hat{\boldsymbol{z}}^{(1)}\|^2 \hat{z}_m^{(2)} - \lambda_m \hat{z}_m^{(1)} \left( \sum_k \lambda_k \hat{z}_k^{(1)} \hat{z}_k^{(2)} \right) \right]$$

$$= -\frac{\lambda_m}{\|D\hat{\boldsymbol{z}}^{(1)}\|^3 \|\hat{\boldsymbol{z}}^{(2)}\|} \left[ \left( \sum_k \lambda_k^2 (\hat{z}_k^{(1)})^2 \right) \hat{z}_m^{(2)} - \lambda_m \hat{z}_m^{(1)} \left( \sum_k \lambda_k \hat{z}_k^{(1)} \hat{z}_k^{(2)} \right) \right]$$

$$= -\frac{\lambda_m}{\|D\hat{\boldsymbol{z}}^{(1)}\|^3 \|\hat{\boldsymbol{z}}^{(2)}\|} \sum_{k\neq m} \lambda_k \left( \lambda_k (\hat{z}_k^{(1)})^2 \hat{z}_m^{(2)} - \lambda_m \hat{z}_m^{(1)} \hat{z}_k^{(1)} \hat{z}_k^{(2)} \right)$$

$$\Rightarrow \frac{\mathrm{d}\hat{z}_m^{(1)}}{\mathrm{d}t} = -\eta \frac{\partial \mathcal{L}}{\partial \hat{z}_m^{(1)}}$$

$$= \frac{\eta \lambda_m}{\|D\hat{\boldsymbol{z}}^{(1)}\|^3 \|\hat{\boldsymbol{z}}^{(2)}\|} \sum_{k\neq m} \lambda_k \left( \lambda_k (\hat{z}_k^{(1)})^2 \hat{z}_m^{(2)} - \lambda_m \hat{z}_m^{(1)} \hat{z}_k^{(1)} \hat{z}_k^{(2)} \right) \quad ,$$

proving Eq. (10). Assuming sufficiently small augmentations, $\hat{z}_k^{(1)}$ and $\hat{z}_k^{(2)}$ carry the same sign, and the net sign of both terms inside the parenthesis is fully determined by $\gamma_m \equiv \mathrm{sign}(\hat{z}_m^{(1)})$. Hence, we may write:

$$\frac{\mathrm{d}\hat{z}_m^{(1)}}{\mathrm{d}t} = \frac{\eta \lambda_m \gamma_m}{\|D\hat{\boldsymbol{z}}^{(1)}\|^3 \|\hat{\boldsymbol{z}}^{(2)}\|} \sum_{k\neq m} \left( \lambda_k^2 (\hat{z}_k^{(1)})^2 |\hat{z}_m^{(2)}| - \lambda_m \lambda_k |\hat{z}_m^{(1)}| |\hat{z}_k^{(1)}| |\hat{z}_k^{(2)}| \right) \quad .$$

It is useful to separate out $\gamma_m$ in this manner because every other term in the expression is now non-negative. Then $\mathrm{sign}(\gamma_m \cdot \frac{\mathrm{d}\hat{z}}{\mathrm{d}t}) = \mathrm{sign}(\hat{z}_m \cdot \frac{\mathrm{d}\hat{z}_m}{\mathrm{d}t})$ tells us whether $\hat{z}_m$ tends to increase or decrease in magnitude, as we have argued in the main text.

**Asymptotic analysis.** To get a handle on how the different eigenvalues influence each other, we consider two important limiting cases. First, we consider the asymptotic regime dominated by one eigenvalue, and show that it tends towards a more symmetric solution in which the gap between different eigenvalues decreases. Second, we derive asymptotic expressions for the near-uniform regime in which all eigenvalues are comparable in size and show that this solution tends toward the uniform solution (cf. Eq. (11)).

To facilitate our analysis, we define each mode's relative contribution $\chi_m \equiv \frac{|\hat{z}_m|}{\|\hat{z}\|}$ and evaluate Eq. (10) taking the expectation value over augmentations:

$$\mathbb{E}\left[\frac{\mathrm{d}\hat{z}_m^{(1)}}{\mathrm{d}t}\right] = \eta\lambda_m \sum_{k\neq m}\left(\lambda_k^2 \cdot \mathbb{E}\left[\frac{(\hat{z}_k^{(1)})^2 \hat{z}_m^{(2)}}{\|D\hat{z}^{(1)}\|^3 \|\hat{z}^{(2)}\|}\right] - \lambda_m\lambda_k \cdot \mathbb{E}\left[\frac{\hat{z}_m^{(1)}\hat{z}_k^{(1)}\hat{z}_k^{(2)}}{\|D\hat{z}^{(1)}\|^3\|\hat{z}^{(2)}\|}\right]\right)$$

$$\frac{\mathrm{d}\hat{z}_m}{\mathrm{d}t} = \eta\lambda_m \sum_{k\neq m}\left(\lambda_k^2 \cdot \mathbb{E}\left[\frac{\hat{z}_k^2}{\|D\hat{z}\|^3}\right]\cdot\mathbb{E}\left[\frac{\hat{z}_m}{\|\hat{z}\|}\right] - \lambda_m\lambda_k \cdot \mathbb{E}\left[\frac{\hat{z}_m\hat{z}_k}{\|D\hat{z}\|^3}\right]\cdot\mathbb{E}\left[\frac{\hat{z}_k}{\|\hat{z}\|}\right]\right)$$

$$= \eta\lambda_m\gamma_m \sum_{k\neq m}\left(\lambda_k^2 \cdot \mathbb{E}\left[\chi_k^2\frac{\|\hat{z}\|^2}{\|D\hat{z}\|^3}\right]\cdot\mathbb{E}\left[\chi_m\right] - \lambda_m\lambda_k \cdot \mathbb{E}\left[\chi_m\chi_k\frac{\|\hat{z}\|^2}{\|D\hat{z}\|^3}\right]\cdot\mathbb{E}\left[\chi_k\right]\right).$$

$$(22)$$

In the second equality we used the fact that the expectation value taken over augmentations is conditioned on the input sample, which makes them conditionally independent.

**One dominant eigenvalue.** First, we consider the low-rank regime in which one eigenvalue dominates. Without loss of generality, we assume $\lambda_1 \gg \lambda_k \; \forall k \neq 1$. We then have:

$$\chi_1 \sim 1$$
$$\chi_k \sim \epsilon \quad (0 < \epsilon \ll 1) \quad \forall \quad k \neq 1$$

Plugging these values into Eq. (22) gives the following dynamics for the dominant eigenmode:

$$\frac{\mathrm{d}\hat{z}_1}{\mathrm{d}t} \approx \eta\lambda_1\gamma_1 \sum_{k\neq 1}\left(\lambda_k^2 \cdot \mathbb{E}\left[\epsilon^2\frac{\|\hat{z}\|^2}{\|D\hat{z}\|^3}\right]\cdot\mathbb{E}\left[1\right] - \lambda_1\lambda_k \cdot \mathbb{E}\left[\epsilon\frac{\|\hat{z}\|^2}{\|D\hat{z}\|^3}\right]\mathbb{E}\left[\epsilon\right]\right)$$

$$= \eta\lambda_1\gamma_1 \sum_{k\neq 1}\left(\lambda_k^2\epsilon^2 \cdot \mathbb{E}\left[\frac{\|\hat{z}\|^2}{\|D\hat{z}\|^3}\right]\cdot\mathbb{E}\left[1\right] - \lambda_1\lambda_k\epsilon^2 \cdot \mathbb{E}\left[\frac{\|\hat{z}\|^2}{\|D\hat{z}\|^3}\right]\right)$$

$$= \eta\lambda_1\gamma_1\epsilon^2\mathbb{E}\left[\frac{\|\hat{z}\|^2}{\|D\hat{z}\|^3}\right]\sum_{k\neq 1}\lambda_k\left(\lambda_k - \lambda_1\right) \quad .$$

These updates are always opposite in sign to the representation component, which corresponds to decaying dynamics for the leading eigenmode because $\gamma_1\frac{\mathrm{d}\hat{z}_1}{\mathrm{d}t} < 0$.

For all other modes we have:

$$\frac{\mathrm{d}\hat{z}_{m\neq 1}}{\mathrm{d}t} \approx \eta\lambda_m\gamma_m \sum_{k\notin\{m,1\}}\left(\lambda_k^2\epsilon^2 \cdot \mathbb{E}\left[\frac{\|\hat{z}\|^2}{\|D\hat{z}\|^3}\right]\cdot\mathbb{E}\left[\epsilon\right] - \lambda_m\lambda_k\epsilon^2 \cdot \mathbb{E}\left[\frac{\|\hat{z}\|^2}{\|D\hat{z}\|^3}\right]\cdot\mathbb{E}\left[\epsilon\right]\right)$$

$$+ \eta\lambda_m\gamma_m\left(\lambda_1^2 \cdot \mathbb{E}\left[\frac{\|\hat{z}\|^2}{\|D\hat{z}\|^3}\right]\cdot\mathbb{E}\left[\epsilon\right] - \lambda_m\lambda_1\epsilon \cdot \mathbb{E}\left[\frac{\|\hat{z}\|^2}{\|D\hat{z}\|^3}\right]\cdot\mathbb{E}\left[1\right]\right)$$

$$= \eta\lambda_m\gamma_m\epsilon \cdot \mathbb{E}\left[\frac{\|\hat{z}\|^2}{\|D\hat{z}\|^3}\right]\left(\lambda_1(\lambda_1 - \lambda_m) + \epsilon^2\sum_{k\notin\{m,1\}}\lambda_k(\lambda_k - \lambda_m)\right)$$

$$\approx \eta\lambda_m\gamma_m\lambda_1\epsilon \cdot \mathbb{E}\left[\frac{\|\hat{z}\|^2}{\|D\hat{z}\|^3}\right](\lambda_1 - \lambda_m) \quad ,$$

so that $\gamma_m\frac{\mathrm{d}\hat{z}_m}{\mathrm{d}t} > 0$, i.e, the updates have the *same* sign as the representation component, which corresponds to growth dynamics. In other words: The dominant eigenvalue "pulls all the other eigenvalues up," a form of implicit cooperation between the eigenmodes. We also note that the non-dominant eigenmodes increase at a rate proportional to $\epsilon$, whereas the dominant eigenmode decreases at a slower rate proportional to $\epsilon^2$. Thus, for sensible initializations with at least one large and many small eigenvalues, the modes will tend toward an equilibrium at some non-zero intermediate value, without a dominant mode. Next we study this other limiting case in which all eigenvalues are of similar size.

**Near-uniform regime.** To study the dynamics in a near-uniform regime, we note that all $\chi_m$ are of order $\mathcal{O}(1)$ in $\hat{z}_m$, whereas the eigenvalues $\lambda_m$ are of order $\mathcal{O}(\hat{z}_m^2)$. In this setting, the effect of the eigenvalue terms $\lambda_m$ on the dynamics is stronger than the $\chi_m$ terms which are bounded between 0 and 1. With a sufficiently high-dimensional representation, all $\chi_m$ terms will be centered around $1/\sqrt{M}$. Based on these observations, we may make the simplifying assumption that the contributions are all approximately equal, i.e, $\chi_i = \chi$ for all $i$. Substituting this value in Eq (22) gives:

$$\frac{\mathrm{d}\hat{z}_m}{\mathrm{d}t} = \eta\lambda_m\gamma_m \cdot \mathbb{E}\left[\chi^2 \frac{\|\hat{\boldsymbol{z}}\|^2}{\|D\hat{\boldsymbol{z}}\|^3}\right] \cdot \mathbb{E}\left[\chi\right] \cdot \sum_{k \neq m} \lambda_k\left(\lambda_k - \lambda_m\right) \quad . \tag{23}$$

Finally, substituting for $\chi$, which by assumption are all approximately equal:

$$\chi \approx \chi_m = \frac{|\hat{z}_m|}{\|\hat{\boldsymbol{z}}\|} \quad ,$$

and absorbing back the sign from $\gamma_m$, we obtain the approximate dynamics in Eq (11):

$$\frac{\mathrm{d}\hat{z}_m}{\mathrm{d}t} \approx \eta\lambda_m \cdot \mathbb{E}\left[\frac{\hat{z}_m^2}{\|D\hat{\boldsymbol{z}}\|^3}\right] \cdot \mathbb{E}\left[\frac{\hat{z}_m}{\|\hat{\boldsymbol{z}}\|}\right] \cdot \sum_{k \neq m} \lambda_k\left(\lambda_k - \lambda_m\right)$$

$\square$

# C   Derivation of idealized learning dynamics for different loss variations

## C.1   Removing the stop-grad from the Euclidean loss $\mathcal{L}^{\text{euc}}$

Omitting the stop-grad operator from $\mathcal{L}^{\text{euc}}$ gives:

$$
\begin{aligned}
\mathcal{L}^{\text{euc}}_{\text{noSG}} &= \frac{1}{2}\|W_{\text{P}}\boldsymbol{z}^{(1)} - \boldsymbol{z}^{(2)}\|^2 \\
&= \frac{1}{2}\sum_m^M |\lambda_m \hat{z}_m^{(1)} - \hat{z}_m^{(2)}|^2 \quad .
\end{aligned}
$$

Tracing the steps to prove Theorem 1 and assuming Gaussian i.i.d inputs for a linear network, we write:

$$
\begin{aligned}
\frac{\partial \mathcal{L}^{\text{euc}}_{\text{noSG}}}{\partial \hat{z}_m} &= \frac{\partial \mathcal{L}^{\text{euc}}_{\text{noSG}}}{\partial \hat{z}_m^{(1)}} + \frac{\partial \mathcal{L}^{\text{euc}}_{\text{noSG}}}{\partial \hat{z}_m^{(2)}} \\
&= \left(\lambda_m \hat{z}_m^{(1)} - \hat{z}_m^{(2)}\right)\lambda_m - \left(\lambda_m \hat{z}_m^{(1)} - \hat{z}_m^{(2)}\right) \\
&= \left(\lambda_m \hat{z}_m^{(1)} - \hat{z}_m^{(2)}\right)(\lambda_m - 1) \\
\Rightarrow \frac{\mathrm{d}\hat{z}_m}{\mathrm{d}t} &= -\eta \mathbb{E}\left[\frac{\partial \mathcal{L}^{\text{euc}}_{\text{noSG}}}{\partial \hat{z}_m}\right] \\
&= -\eta\left(\lambda_m \mathbb{E}[\hat{z}_m^{(1)}] - \mathbb{E}[\hat{z}_m^{(2)}]\right)(\lambda_m - 1) \\
&= -\eta\left(1 - \lambda_m\right)^2 \hat{z}_m \quad ,
\end{aligned}
$$

which results in decaying representations and thus collapse.

## C.2   Removing the stop-grad from the cosine loss $\mathcal{L}^{\text{cos}}$

Following the same arguments as above, omitting the stop-grad operator from $\mathcal{L}^{\text{cos}}$ gives:

$$
\begin{aligned}
\mathcal{L}^{\text{cos}}_{\text{noSG}} &= -\frac{\left(W_{\text{P}}\boldsymbol{z}^{(1)}\right)^\top \boldsymbol{z}^{(2)}}{\|W_{\text{P}}\boldsymbol{z}^{(1)}\|\|\boldsymbol{z}^{(2)}\|} \\
\Rightarrow \frac{\partial \mathcal{L}^{\text{cos}}_{\text{noSG}}}{\partial \hat{z}_m} &= \frac{-\lambda_m}{\|D\hat{\boldsymbol{z}}^{(1)}\|^3\|\hat{\boldsymbol{z}}^{(2)}\|}\sum_{k\neq m}\left(\lambda_k^2(\hat{z}_k^{(1)})^2\hat{z}_m^{(2)} - \lambda_m\lambda_k\hat{z}_m^{(1)}\hat{z}_k^{(1)}\hat{z}_k^{(2)} + \lambda_k^2\lambda_m(\hat{z}_k^{(1)})^3 - \lambda_k\hat{z}_m^{(2)}\hat{z}_k^{(2)}\hat{z}_k^{(1)}\right) \\
&\quad + \frac{-\lambda_m}{\|D\hat{\boldsymbol{z}}^{(1)}\|^3\|\hat{\boldsymbol{z}}^{(2)}\|}\left(\lambda_m^3(\hat{z}_m^{(1)})^3 - \lambda_m(\hat{z}_m^{(2)})^2\hat{z}_m^{(1)}\right) \quad ,
\end{aligned}
$$

so that, when taking the expectation value over augmentations, the dynamics follow:

$$
\begin{aligned}
\frac{\mathrm{d}\hat{z}_m}{\mathrm{d}t} &= -\eta \mathbb{E}\left[\frac{\partial \mathcal{L}^{\text{cos}}_{\text{noSG}}}{\partial \hat{z}_m}\right] \\
&= \eta\lambda_m\gamma_m\sum_{k\neq m}\lambda_k\left(\lambda_k \cdot \mathbb{E}\left[\chi_k^2\frac{\|\hat{\boldsymbol{z}}\|^2}{\|D\hat{\boldsymbol{z}}\|^3}\right]\cdot\mathbb{E}\left[\chi_m\right] - \lambda_m\cdot\mathbb{E}\left[\chi_m\chi_k\frac{\|\hat{\boldsymbol{z}}\|^2}{\|D\hat{\boldsymbol{z}}\|^3}\right]\cdot\mathbb{E}\left[\chi_k\right]\right) \\
&\quad + \eta\lambda_m\gamma_m\sum_{k\neq m}\lambda_k\left(\lambda_m\lambda_k\mathbb{E}\left[\chi_k^3\frac{\|\hat{\boldsymbol{z}}\|^3}{\|D\hat{\boldsymbol{z}}\|^3}\right]\cdot\mathbb{E}\left[\frac{1}{\|\hat{\boldsymbol{z}}\|}\right] - \mathbb{E}\left[\chi_k\frac{\|\hat{\boldsymbol{z}}\|}{\|D\hat{\boldsymbol{z}}\|^3}\right]\cdot\mathbb{E}\left[\chi_m\chi_k\|\hat{\boldsymbol{z}}\|\right]\right) \\
&\quad + \eta\lambda_m^2\gamma_m\left(\lambda_m^2\mathbb{E}\left[\chi_m^3\frac{\|\hat{\boldsymbol{z}}\|^3}{\|D\hat{\boldsymbol{z}}\|^3}\right]\cdot\mathbb{E}\left[\frac{1}{\|\hat{\boldsymbol{z}}\|}\right] - \mathbb{E}\left[\chi_m\frac{\|\hat{\boldsymbol{z}}\|}{\|D\hat{\boldsymbol{z}}\|^3}\right]\cdot\mathbb{E}\left[\chi_m^2\|\hat{\boldsymbol{z}}\|\right]\right) \quad .
\end{aligned}
$$

In the asymptotic regime with dominant eigenvalue $\lambda_1$, we get the dynamics:

$$\frac{\mathrm{d}\hat{z}_1}{\mathrm{d}t} = \eta\lambda_1\gamma_1 \sum_{k\neq m} \lambda_k \left( \lambda_k\epsilon^2\mathbb{E}\left[\frac{\|\hat{z}\|^2}{\|D\hat{z}\|^3}\right] - \lambda_m\epsilon^2\mathbb{E}\left[\frac{\|\hat{z}\|^2}{\|D\hat{z}\|^3}\right] \right)$$

$$+ \eta\lambda_1\gamma_1 \sum_{k\neq m} \lambda_k \left( \lambda_1\lambda_m\epsilon^3\mathbb{E}\left[\frac{\|\hat{z}\|^3}{\|D\hat{z}\|^3}\right]\cdot\mathbb{E}\left[\frac{1}{\|\hat{z}\|}\right] - \epsilon^2\mathbb{E}\left[\frac{\|\hat{z}\|}{\|D\hat{z}\|^3}\right]\cdot\mathbb{E}\left[\|\hat{z}\|\right] \right)$$

$$+ \eta\lambda_1^2\gamma_1 \left( \lambda_1^2\cdot\mathbb{E}\left[\frac{\|\hat{z}\|^3}{\|D\hat{z}\|^3}\right]\cdot\mathbb{E}\left[\frac{1}{\|\hat{z}\|}\right] - \mathbb{E}\left[\frac{\|\hat{z}\|}{\|D\hat{z}\|^3}\right]\cdot\mathbb{E}\left[\|\hat{z}\|\right] \right)$$

$$\approx \eta\lambda_1^4\gamma_1\cdot\mathbb{E}\left[\frac{\|\hat{z}\|^3}{\|D\hat{z}\|^3}\right]\cdot\mathbb{E}\left[\frac{1}{\|\hat{z}\|}\right]$$

$$\frac{\mathrm{d}\hat{z}_{m\neq1}}{\mathrm{d}t} = \eta\lambda_m\gamma_m \sum_{k\notin\{m,1\}} \lambda_k \left( \lambda_k\epsilon^3\mathbb{E}\left[\frac{\|\hat{z}\|^2}{\|D\hat{z}\|^3}\right] - \lambda_m\epsilon^3\mathbb{E}\left[\frac{\|\hat{z}\|^2}{\|D\hat{z}\|^3}\right] \right)$$

$$+ \eta\lambda_m\gamma_m\lambda_1 \left( \lambda_1\epsilon\mathbb{E}\left[\frac{\|\hat{z}\|^2}{\|D\hat{z}\|^3}\right] - \lambda_m\epsilon\mathbb{E}\left[\frac{\|\hat{z}\|^2}{\|D\hat{z}\|^3}\right] \right)$$

$$+ \eta\lambda_m\gamma_m \sum_{k\notin\{m,1\}} \lambda_k \left( \lambda_m\lambda_k\epsilon^3\mathbb{E}\left[\frac{\|\hat{z}\|^3}{\|D\hat{z}\|^3}\right]\cdot\mathbb{E}\left[\frac{1}{\|\hat{z}\|}\right] - \epsilon^3\mathbb{E}\left[\frac{\|\hat{z}\|}{\|D\hat{z}\|^3}\right]\cdot\mathbb{E}\left[\|\hat{z}\|\right] \right)$$

$$+ \eta\lambda_m\gamma_m\lambda_1 \left( \lambda_m\lambda_1\mathbb{E}\left[\frac{\|\hat{z}\|^3}{\|D\hat{z}\|^3}\right]\cdot\mathbb{E}\left[\frac{1}{\|\hat{z}\|}\right] - \epsilon\mathbb{E}\left[\frac{\|\hat{z}\|}{\|D\hat{z}\|^3}\right]\cdot\mathbb{E}\left[\|\hat{z}\|\right] \right)$$

$$+ \eta\lambda_m^2\gamma_m \left( \lambda_m^2\epsilon^3\mathbb{E}\left[\frac{\|\hat{z}\|^3}{\|D\hat{z}\|^3}\right]\cdot\mathbb{E}\left[\frac{1}{\|\hat{z}\|}\right] - \epsilon^3\mathbb{E}\left[\frac{\|\hat{z}\|}{\|D\hat{z}\|^3}\right]\cdot\mathbb{E}\left[\|\hat{z}\|\right] \right)$$

$$\approx \eta\lambda_m^2\gamma_m\lambda_1^2\cdot\mathbb{E}\left[\frac{\|\hat{z}\|^3}{\|D\hat{z}\|^3}\right]\cdot\mathbb{E}\left[\frac{1}{\|\hat{z}\|}\right],$$

Thus, all eigenmodes diverge because $\gamma_m\frac{\mathrm{d}\hat{z}_m}{\mathrm{d}t} > 0$.

Similarly, we find divergent dynamics when starting in the near-uniform regime:

$$\frac{\mathrm{d}\hat{z}_m}{\mathrm{d}t} = \eta\lambda_m\gamma_m\mathbb{E}\left[\chi^2\frac{\|\hat{z}\|^2}{\|D\hat{z}\|^3}\right]\cdot\mathbb{E}\left[\chi\right]\sum_{k\neq m}\lambda_k(\lambda_k - \lambda_m)$$

$$+ \eta\lambda_m^2\gamma_m\mathbb{E}\left[\chi^3\frac{\|\hat{z}\|^3}{\|D\hat{z}\|^3}\right]\cdot\mathbb{E}\left[\frac{1}{\|\hat{z}\|}\right]\sum_k\lambda_k^2$$

$$- \eta\lambda_m\gamma_m\mathbb{E}\left[\chi\frac{\|\hat{z}\|}{\|D\hat{z}\|^3}\right]\cdot\mathbb{E}\left[\chi^2\|\hat{z}\|\right]\sum_k\lambda_k$$

$$\approx \eta\lambda_m^2\gamma_m\mathbb{E}\left[\chi^3\frac{\|\hat{z}\|^3}{\|D\hat{z}\|^3}\right]\cdot\mathbb{E}\left[\frac{1}{\|\hat{z}\|}\right]\sum_k\lambda_k^2 \quad,$$

selecting the terms with the highest power in the eigenvalues.

Thus, omission of stop-grad precludes successful representation learning for both the Euclidean and the cosine loss, but due to different mechanisms. Euclidean loss yields collapse, whereas the cosine loss succumbs to run-away activity.

## C.3  Removing the predictor from the Euclidean loss $\mathcal{L}^{\mathrm{euc}}$

To analyze the representational dynamics in the absence of the predictor network, we consider $\mathcal{L}^{\mathrm{euc}}_{\mathrm{noPred}}$:

$$\mathcal{L}^{\mathrm{euc}}_{\mathrm{noPred}} = \frac{1}{2}\|z^{(1)} - \mathrm{SG}(z^{(2)})\|^2$$

$$= \frac{1}{2}\sum_m^M |\hat{z}_m^{(1)} - \mathrm{SG}(\hat{z}_m^{(2)})|^2 \quad.$$

The dynamics resulting from this loss function are a special case of the dynamics derived in Theorem 1 with all the eigenvalues equal to one ($\lambda_k = 1$). In particular Eq. (8) becomes:

$$\frac{d\hat{z}_m^{(1)}}{dt} = -\eta \frac{\partial \mathcal{L}_{\text{noPred}}^{\text{euc}}}{\partial \hat{z}_m^{(1)}}(t) = \eta \left( \hat{z}_m^{(2)} - \hat{z}_m^{(1)} \right),$$

which evaluates to 0 under expectation over augmentations. Hence there is no learning without the predictor.

## C.4 Removing the predictor from the cosine loss $\mathcal{L}^{\cos}$

Similarly, when we remove the predictor from $\mathcal{L}^{\cos}$ it yields:

$$\mathcal{L}_{\text{noPred}}^{\cos} = -\sum_m^M \frac{\hat{z}_m^{(1)} \text{SG}(\hat{z}_m^{(2)})}{\|\hat{\boldsymbol{z}}^{(1)}\| \|\text{SG}(\hat{\boldsymbol{z}}^{(2)})\|},$$

so that Eq. (10) becomes:

$$\frac{d\hat{z}_m^{(1)}}{dt} = \frac{\eta}{\|\hat{\boldsymbol{z}}^{(1)}\|^3 \|\hat{\boldsymbol{z}}^{(2)}\|} \sum_{k \neq m} \left( (\hat{z}_k^{(1)})^2 \hat{z}_m^{(2)} - \hat{z}_m^{(1)} \hat{z}_k^{(1)} \hat{z}_k^{(2)} \right). \tag{24}$$

Here, the near-uniform approximation (Eq. (11)) of ignoring the differences in $\chi$ between different eigenmodes is not valid. This is because the $\lambda$-terms are no longer present, and the effects of the $\chi$-terms on the dynamics cannot be treated as negligible. In particular, setting $W_{\text{P}} = I$ in the dynamics derived in Theorem 2 would yield zero dynamics. However taking the expectation of Eq. (24) over augmentations yields the non-zero dynamics:

$$\frac{d\hat{z}_m}{dt} = \eta \sum_{k \neq m} \left( \mathbb{E}\left[ \frac{\hat{z}_k^2}{\|\hat{\boldsymbol{z}}\|^3} \right] \mathbb{E}\left[ \frac{\hat{z}_m}{\|\hat{\boldsymbol{z}}\|} \right] - \mathbb{E}\left[ \frac{\hat{z}_m \hat{z}_k}{\|\hat{\boldsymbol{z}}\|^3} \right] \mathbb{E}\left[ \frac{\hat{z}_k}{\|\hat{\boldsymbol{z}}\|} \right] \right), \tag{25}$$

which consists of terms of order $\mathcal{O}\left( \frac{1}{\hat{z}_m} \right)$ and $\mathcal{O}(1)$ in $\hat{z}_m$, hinting at slower dynamics compared to Eq. (10). To study these granular effects, we would need to explicitly model the effect of the augmentations, for which we do not have a good statistical model. In lieu of deriving these dynamics analytically, we make an observation which restricts the possible dynamical behavior. Specifically, the sum of the eigenvalues remains constant throughout training. To show this, we begin by writing out the expression for the derivative over time of the sum of all the eigenvalues:

$$\frac{d}{dt} \sum_m \lambda_m = \sum_m \frac{d\lambda_m}{dt} = \sum_m \frac{d}{dt} \mathbb{E}_{\text{data}}\left[ \hat{z}_m^2 \right]$$

$$= \mathbb{E}_{\text{data}}\left[ \sum_m \frac{d\hat{z}_m^2}{dt} \right]$$

$$= \mathbb{E}_{\text{data}}\left[ \sum_m \hat{z}_m \frac{d\hat{z}_m}{dt} \right].$$

We can derive the term inside the expectation by adding up the dynamics given by Eq. (25) for all the different eigenmodes:

$$\hat{z}_m^{(1)} \frac{d\hat{z}_m^{(1)}}{dt} = \frac{\eta}{\|\hat{\boldsymbol{z}}^{(1)}\|^3 \|\hat{\boldsymbol{z}}^{(2)}\|} \sum_{k \neq m} \left( (\hat{z}_k^{(1)})^2 \hat{z}_m^{(1)} \hat{z}_m^{(2)} - (\hat{z}_m^{(1)})^2 \hat{z}_k^{(1)} \hat{z}_k^{(2)} \right)$$

$$\implies \sum_m \hat{z}_m^{(1)} \frac{d\hat{z}_m^{(1)}}{dt} = \frac{\eta}{\|\hat{\boldsymbol{z}}^{(1)}\|^3 \|\hat{\boldsymbol{z}}^{(2)}\|} \sum_m \sum_{k \neq m} \left( (\hat{z}_k^{(1)})^2 \hat{z}_m^{(1)} \hat{z}_m^{(2)} - (\hat{z}_m^{(1)})^2 \hat{z}_k^{(1)} \hat{z}_k^{(2)} \right)$$

$$= 0$$

$$\implies \mathbb{E}_{\text{data}}\left[ \mathbb{E}_{\text{aug}}\left[ \sum_m \hat{z}_m^{(1)} \frac{d\hat{z}_m^{(1)}}{dt} \right] \right] = \mathbb{E}_{\text{data}}\left[ \sum_m \hat{z}_m \frac{d\hat{z}_m}{dt} \right] = 0,$$

proving that $\frac{d}{dt} \sum_m \lambda_m = 0$, i.e, the sum of the eigenvalues is conserved. This precludes collapsing dynamics where all eigenvalues go to zero as well as diverging dynamics where at least one eigenvalue diverges.

## C.5 Isotropic losses for equalized convergence rates

In Expressions (9) and (11) we see that the overall learning dynamics have a quadratic dependence on the eigenvalues with a root near collapsed solutions, which causes these modes to learn slower. We reasoned that this anisotropy could be detrimental for learning. To address this issue, we sought to derive alternative loss functions that encourage isotropic learning dynamics for all modes.

### C.5.1 Euclidean IsoLoss.

We start by deriving an IsoLoss function for the Euclidean case $\mathcal{L}^{\text{euc}}$. To avoid the unwanted quadratic dependence, we first note that we would like to arrive at the following expression for the dynamics:

$$\frac{\mathrm{d}\hat{z}_m}{\mathrm{d}t} = \eta\left(1 - \lambda_m\right)\hat{z}_m \quad .$$

By recalling the Euclidean loss and corresponding dynamics:

$$\mathcal{L}^{\text{euc}} = \tfrac{1}{2}\sum_m^M |\lambda_m\hat{z}_m^{(1)} - \text{SG}(\hat{z}_m^{(2)})|^2 \Rightarrow \frac{\mathrm{d}\hat{z}_m}{\mathrm{d}t} = \eta\lambda_m\left(1 - \lambda_m\right)\hat{z}_m \quad ,$$

we note that the leading $\lambda_m$ term has no influence on the overall sign of the dynamics, and is introduced by the second step in the chain rule:

$$\frac{\partial\mathcal{L}^{\text{euc}}}{\partial\hat{z}_m^{(1)}} = (\lambda_m\hat{z}^{(1)} - \hat{z}^{(2)}) \cdot \frac{\partial}{\partial\hat{z}_m^{(1)}}(\lambda_m\hat{z}^{(1)} - \hat{z}^{(2)}) \quad .$$

Based on this realization we see that this second step needs to be modified. To that end, we start with the desired derivative:

$$\frac{\partial\mathcal{L}^{\text{euc}}_{\text{iso}}}{\partial\hat{z}_m^{(1)}} = (\lambda_m\hat{z}^{(1)} - \hat{z}^{(2)}) \cdot \frac{\partial}{\partial\hat{z}_m^{(1)}}(\hat{z}^{(1)} - \hat{z}^{(2)}) \quad ,$$

and see that several loss functions are possible. The one we have reported in Eq. (15) we derived by applying an appropriate stop-grad while integrating:

$$\frac{\partial\mathcal{L}^{\text{euc}}_{\text{iso}}}{\partial\hat{z}_m^{(1)}} = (\hat{z}_m^{(1)} + \lambda_m\hat{z}^{(1)} - \hat{z}^{(2)} - \hat{z}_m^{(1)}) \cdot \frac{\partial}{\partial\hat{z}_m^{(1)}}(\hat{z}^{(1)} - \hat{z}^{(2)}) \quad .$$

to give:

$$\mathcal{L}^{\text{euc}}_{\text{iso}} = \tfrac{1}{2}\sum_m^M |\hat{z}_m^{(1)} - \text{SG}(\hat{z}_m^{(2)} + \hat{z}_m^{(1)} - \lambda_m\hat{z}_m^{(1)})|^2$$

Another alternative loss with the same desired isotropic learning dynamics, but using a different placement of the stop-gradient operators, is given by:

$$\mathcal{L}^{\text{euc}}_{\text{iso}} = \sum_m^M \text{SG}\left(\lambda_m\hat{z}_m^{(1)} - \hat{z}_m^{(2)}\right) \cdot \left(\hat{z}_m^{(1)} - \text{SG}(\hat{z}_m^{(2)})\right)$$

### C.5.2 Cosine Similarity IsoLoss.

Since most practical SSL approaches rely on cosine similarity, which suffers from a similar anisotropy of the learning dynamics, we sought to find IsoLosses in this setting. With the same goal as above, we would like to arrive at the dynamics:

$$\frac{\mathrm{d}\hat{z}_m}{\mathrm{d}t} = \eta\frac{\hat{z}_m^{(2)}}{\|D\hat{z}^{(1)}\|\|\hat{z}^{(2)}\|} - \eta\frac{\sum_k \lambda_k\hat{z}_k^{(1)}\hat{z}_k^{(2)}}{\|D\hat{z}^{(1)}\|^3\|\hat{z}^{(2)}\|} \cdot \lambda_m\hat{z}_m^{(1)}$$

starting from the cosine loss and corresponding dynamics:

$$\mathcal{L}^{\text{cos}} = -\sum_m^M \frac{\lambda_m\hat{z}_m^{(1)}\,\text{SG}(\hat{z}_m^{(2)})}{\|D\hat{z}^{(1)}\|\|\text{SG}(\hat{z}^{(2)})\|} \tag{26}$$

$$\Rightarrow \frac{\mathrm{d}\hat{z}_m}{\mathrm{d}t} = \eta\frac{\lambda_m\hat{z}_m^{(2)}}{\|D\hat{z}^{(1)}\|\|\hat{z}^{(2)}\|} - \eta\frac{\sum_k \lambda_k\hat{z}_k^{(1)}\hat{z}_k^{(2)}}{\|D\hat{z}^{(1)}\|^3\|\hat{z}^{(2)}\|} \cdot \lambda_m^2\hat{z}_m^{(1)} \quad . \tag{27}$$

The IsoLoss in this case can be derived by noting how $\lambda_m$ arises in each of the two terms in Eq. (27), and engineering an alternative loss function corresponding to each term separately.

In the first term, $\lambda_m$ arises from the partial derivative of the numerator $\lambda_m \hat{z}_m^{(1)} \mathrm{SG}(\hat{z}_m^{(2)})$ in the original loss (Eq. (26)). This can be remediated by using $\hat{z}_m^{(1)} \mathrm{SG}(\hat{z}_m^{(2)})$ as the numerator instead.

In the second term in Eq. (27), $\lambda_m^2$ arises from the partial derivative of $\|D\hat{\boldsymbol{z}}^{(1)}\| = \sqrt{\sum_k (\lambda_k \hat{z}_m^{(1)})^2}$ in the denominator. We can reduce $\lambda_m^2$ to $\lambda_m$ by instead taking the partial derivative of $\|D^{1/2}\hat{\boldsymbol{z}}^{(1)}\| = \sqrt{\sum_k (\lambda_k^{1/2} \hat{z}_m^{(1)})^2}$.

Putting these insights together, we arrive at the desired partial derivative:

$$
\begin{aligned}
\frac{\partial \mathcal{L}_{\mathrm{iso}}^{\cos}}{\partial \hat{z}_m^{(1)}} &= \frac{-1}{\|D\hat{\boldsymbol{z}}^{(1)}\|\|\hat{\boldsymbol{z}}^{(2)}\|} \cdot \frac{\partial \hat{z}_m^{(1)} \hat{z}_m^{(2)}}{\partial \hat{z}_m^{(1)}} + \frac{\sum_k \lambda_k \hat{z}_k^{(1)} \hat{z}_k^{(2)}}{\|D\hat{\boldsymbol{z}}^{(1)}\|^3 \|\hat{\boldsymbol{z}}^{(2)}\|} \cdot \frac{1}{2} \frac{\partial \lambda_m (\hat{z}_m^{(1)})^2}{\partial \hat{z}_m^{(1)}} \\
&= \frac{-1}{\|D\hat{\boldsymbol{z}}^{(1)}\|\|\hat{\boldsymbol{z}}^{(2)}\|} \cdot \frac{\partial (\hat{\boldsymbol{z}}^{(1)})^\top \hat{\boldsymbol{z}}^{(2)}}{\partial \hat{z}_m^{(1)}} + \frac{\sum_k \lambda_k \hat{z}_k^{(1)} \hat{z}_k^{(2)}}{\|D\hat{\boldsymbol{z}}^{(1)}\|^3 \|\hat{\boldsymbol{z}}^{(2)}\|} \cdot \frac{1}{2} \frac{\partial \|D^{1/2}\hat{\boldsymbol{z}}^{(1)}\|^2}{\partial \hat{z}_m^{(1)}}
\end{aligned} \quad ,
$$

and the integrated IsoLoss in eigenspace:

$$
\mathcal{L}_{\mathrm{iso}}^{\cos} = -(\hat{\boldsymbol{z}}^{(1)})^\top \mathrm{SG}\left( \frac{\hat{\boldsymbol{z}}^{(2)}}{\|D\hat{\boldsymbol{z}}^{(1)}\|\|\hat{\boldsymbol{z}}^{(2)}\|} \right) + \frac{1}{2}\mathrm{SG}\left( \frac{(D\hat{\boldsymbol{z}}^{(1)})^\top \hat{\boldsymbol{z}}^{(2)}}{\|D\hat{\boldsymbol{z}}^{(1)}\|^3 \|\hat{\boldsymbol{z}}^{(2)}\|} \right) \|D^{1/2}\hat{\boldsymbol{z}}^{(1)}\|^2 \quad .
$$

Rotating all terms back to the original space gives the desired IsoLoss for cosine similarity as reported (Eq. (17)):

$$
\mathcal{L}_{\mathrm{iso}} = -(\boldsymbol{z}^{(1)})^\top \mathrm{SG}\left( \frac{\boldsymbol{z}^{(2)}}{\|W_{\mathrm{P}}\boldsymbol{z}^{(1)}\|\|\boldsymbol{z}^{(2)}\|} \right) + \frac{1}{2}\mathrm{SG}\left( \frac{(W_{\mathrm{P}}\boldsymbol{z}^{(1)})^\top \boldsymbol{z}^{(2)}}{\|W_{\mathrm{P}}\boldsymbol{z}^{(1)}\|^3 \|\boldsymbol{z}^{(2)}\|} \right) \|W_{\mathrm{P}}^{1/2}\boldsymbol{z}^{(1)}\|^2 \quad .
$$

# D   Experimental methods

**Self-supervised pretraining.**   We used the CIFAR-10, CIFAR-100 [3], STL-10 [4], and TinyImageNet [5] datasets for self-supervised learning with a ResNet-18 [6] encoder and the SimCLR set of transformations [7]. We also adopted several modifications of ResNet-18 which have been proposed to deal with the low resolution of the images in these datasets [7]. The ResNet modifications comprise using $3 \times 3$ convolutional kernels instead of $7 \times 7$ kernels and skipping the first max-pooling operation. The Solo-learn library [8] also provides specialized sets of augmentations that work well for these datasets, which we adopted as well. The configurations we used for each dataset are summarized in Table 4. We used BatchNorm in the backbone and the projector multi-layer perceptron (MLP) in the hidden layer for all methods. For BYOL, we included BatchNorm also in the hidden layer of the predictor MLP.

We used a projection dimension of 256 for the projection MLP using one hidden layer with 4096 units, and the same architecture for the nonlinear predictor for the BYOL baseline. For networks using EMA target networks, we used the LARS optimizer with learning rate 1.0 whereas for networks without the EMA, we used stochastic gradient descent with momentum 0.9 and learning rate 0.1. Furthermore, we used a warmup period of 10 epochs for the learning rate followed by a cosine decay schedule and a batch size of 256. We also used a weight decay $4 \times 10^{-4}$ for the closed-form predictor models and $10^{-5}$ for the nonlinear predictor models. For evaluation, we removed the projection MLP and used the embeddings at the pooled output of the ResNet convolutional layers following standard practice. For the EMA, we started with $\tau_{\text{base}} = 0.99$ and increased $\tau_{\text{EMA}}$ to 1 with a cosine schedule exactly following the configuration reported in [9]. For DirectPred, we used $\alpha = 0.5$ and $\tau = 0.998$ for the moving average estimate of the correlation matrix updated at every step.

Table 4:

|  | CIFAR-10 | CIFAR-100 | STL-10 | TinyImageNet |
|---|---|---|---|---|
| Resolution | $32 \times 32$ | $32 \times 32$ | $96 \times 96$ | $64 \times 64$ |
| Kernel size | $3 \times 3$ | $3 \times 3$ | $7 \times 7$ | $3 \times 3$ |
| First max-pool | No | No | Yes | Yes |
| Blur | No | No | Yes | No |

**Linear evaluation protocol.**   We reported the held-out classification accuracy on the test sets for CIFAR-10/100 and STL-10, and the validation set for TinyImageNet, after online training of the gradient-isolated linear classifier on each labeled example in the training set during pretraining.

**Compute resources**   All simulations were run on an in-house cluster consisting of 5 nodes with 4 V100 NVIDIA GPUs each, one node with 4 A100 NVIDIA GPUs, and one node with 8 A40 NVIDIA GPUs. Runs on CIFAR-10/100 took about 8 hours each, and the STL-10 and TinyImagenet runs took about 24 hours each.

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
