# OpenReview forum: "Implicit variance regularization in non-contrastive SSL"
_NeurIPS.cc/2023/Conference — NeurIPS 2023 poster_

### Official Review · Reviewer_qXm9 · 2023-07-07

**Soundness:** 3 good
**Presentation:** 2 fair
**Contribution:** 3 good
**Rating:** 6
**Confidence:** 3

**Summary:**

This article builds on the theory of non-constrastive (nc) self-supervised learning (SSL), with methods such as BYOL or SimSiam. Contrary to the existing litterature, they study directly the cosine similarity loss used rather than a Euclidian loss on the eigenspace, using NTK dynamics. They show that collapse is avoided in this case with implicit variance reduction. With their analysis, they find an undesirable anisotropy which they fix with a new family of losses that outperforms the state-of-the-art, especially without needing an Exponiential Moving Average (EMA) target network.

**Strengths:**

The theory provides strong results in the linear framework, explaining the eigenvalues dynamics of both euclidian and cosine losses with and without projectors and stop gradients. The use of the NTK dynamics and the analysis of the cosine similarity is novel to me.

The findings that the eigenvalues affect learning as a learning rate provide a good justification for the new IsoLosses, which gives state-of-the-art results. In particular, the removal of the EMA and the increased dimensionality of the losses ensures their consistency.

**Weaknesses:**

One of the main claims of the article is the analysis of the cosine similarity loss. However in that case, if the dynamics are quite different as claimed by the authors, I find the use of the result of Tian et al. that the predictor eigenspace aligns with the one of the correlation matrix of the representations surprising since it was established with a Euclidian loss. I would have hoped atleast a comment on this, if not an analysis of the alignment as done in Tian et al.

I was awaiting an analysis of the eigenvalues similar to Figure 2 for a real network, and not only a comparison with other losses. As such, the analysis of the IsoLoss feels a bit disconnected from the theoretical analysis. Why do the dimensionality increase? Is it because small eigenvalues do not collapse due to their low learning rates ? Do the eigenvalues evolve during training similarly to the linear case, for BYOL/SimSiam and IsoLoss ? (and their euclidian counterparts)

Two important notions in the theory of NC-SSL are the EMA and the weight decay. Since an important improvement of the method is the removal of the need of the EMA of the target network, I would find natural to mention the effect of the EMA in the dynamic.
Similarly, the results of Wang et al. indicate the importance of the role of weight decay to collapse unwanted eigenvalues. I am surprised that the role of weight decay is not mentioned anywhere, especially despite its use in the implementation.

The formulation of the Isoloss (in particular the cosine similarity one) is very hard to interpret. However, it is logical due to the way it was found.

This is an interesting theoretical article despite some flaws and questions raised, and I am ready to raise my score if my questions are answered.

**Questions:**

If would define $\hat z = U z$ a bit more clearly maybe before Lemma 1.

Equation 3: There is a dimensional issue with the diagonal matrix on the right with the product "$zD$" if I am not mistaken. Shouldn't $D$ be on the left?

Definition 1: I find the different notations often a bit unclear, but most importantly is the sudden usage of the time $t$ index which is not consistent. It is present at line 114 but not at line 111, and appears only on part of the variables. Similarly for lemma 2, where $\hat z_t$ loses the time index again. The use of bold variables is also often inconsistent.

Table 2: I would have expected the inclusion of DirectCopy and of SimSiam in the comparison.

**Limitations:**

The authors address adequately the limits of the theory they established, although they do not talk about the effect of EMA and weight decay in their theory or their hypothesis of DirectPred which was established with an Euclidian loss.

---

> ### Author Rebuttal · Authors · 2023-08-09
>
> Thanks for your questions and suggestions. It resulted in an exciting new figure showing that our theory does apply to general settings. Please find below our point-by-point response
>
> > *“I find the use of the result of Tian et al. that the predictor eigenspace aligns with the one of the correlation matrix of the representations surprising since it was established with a Euclidian loss. I would have hoped at least a comment on this, if not an analysis of the alignment as done in Tian et al.”*
>
> Thanks for making an important point. The predictor eigenspace in fact still aligns with the correlation matrix of representation under the cosine loss. This can be understood with a timescale argument. If the linear predictor is trained until convergence on a faster timescale than the encoder, then the converged predictor is the same regardless of the distance metric used, because both L2 and cosine distances have the same optimal solution for the linear predictor. For a cosine distance metric, an optimal linear reconstructor is still the PCA matrix. We propose to add the following comment in the manuscript where DirectPred is introduced (line 83), as well as an additional Appendix section detailing our reasoning:
>
> *“Although Tian et al derived the optimal predictor in the Euclidean loss case, we show in Appendix that the optimal predictor is the same for the cosine loss”*
>
> > *“Why do the dimensionality increase? Is it because small eigenvalues do not collapse due to their low learning rates ? Do the eigenvalues evolve during training similarly to the linear case, for BYOL/SimSiam and IsoLoss ? (and their euclidean counterparts)”*
>
> Yes, we believe the dimensionality increases because IsoLoss recruits more small eigenvalues. We now have numerical support for this hypothesis which we will add in a new Figure in the revised text showing the eigenvalue evolution during training for the different cases (Rebuttal Fig. 1). Many thanks for this question. Looking into this matter revealed quite different eigenvalue dynamics for direct predictor approaches and nonlinear predictor networks like BYOL. This observation raises a host of new questions that we will communicate in the final paper and whose origins we will study in more detail in the future.
>
> > *“Since an important improvement of the method is the removal of the need of the EMA of the target network, I would find natural to mention the effect of the EMA in the dynamic.”*
>
> Thanks for raising this point. We should have highlighted it in the submission. It is true that our theory does not directly include the effect of the EMA. However, the need for an EMA in DirectPred (Tian et al) was justified by its “automatic curriculum” effect on small eigenvalues. Since IsoLoss provides an alternative way to accelerate learning of those small eigenvalues, it makes sense that it is robust to removal. We will add this remark in the revised version of the manuscript.
>
> > *“I am surprised that the role of weight decay is not mentioned anywhere, especially despite its use in the implementation.”*
>
> Thank you for pointing this out. The work by Wang et. al. indeed is relevant to our discussion on eigenvalue recruitment. However, their work is largely orthogonal to ours since they discuss the threshold set by weight decay on across-augmentation variance, whereas we are primarily concerned with across-data variances. We do not include it in our analysis to avoid complication, but use it in our simulations in order to get good performance. We will include this in our discussion in the updated version. Revisiting this paper also suggested to us an interesting experiment, wherein lower weight decay could increase the number of eigenvalues potentially recruited for all losses and might complement the effect of IsoLoss. We have not looked carefully into this matter, but we do provide some initial plots in Rebuttal Fig. 1c showing the effect of weight decay on the dynamics. We will study this further in future work.
>
> > *“If would define $\hat{z} = Uz$ a bit more clearly maybe before Lemma 1.”*
>
> Thank you for the suggestion. It would indeed add to the clarity of the paper. We will include this change in the final version.
>
> > *“Equation 3: There is a dimensional issue with the diagonal matrix on the right with the product "$zD$" if I am not mistaken. Shouldn't $D$ be on the left?”*
>
> Thank you for reading carefully and pointing out this mistake. It should indeed be on the left to be consistent with our column-vector convention, and will be corrected. Since D is a diagonal square matrix, its position doesn’t affect the final product and the rest of our analysis.
>
> > *“Definition 1: I find the different notations often a bit unclear, but most importantly is the sudden usage of the time  index which is not consistent. It is present at line 114 but not at line 111, and appears only on part of the variables. Similarly for lemma 2, where $\hat{z}_t$ loses the time index again. The use of bold variables is also often inconsistent.”*
>
> Thank you for your careful feedback. Apologies for the mistakes. Further, we agree that the time index is not consistently used. Specifically, it should always be included as a subscript on the representations z starting from line 111 until line 140, after which we explicitly omit it for clarity. We will also double-check our use of boldsymbol for all vector quantities. These inaccuracies and mistakes will be corrected.
>
> > *“Table 2: I would have expected the inclusion of DirectCopy and of SimSiam in the comparison.”*
>
> These would be useful inclusions. We have included single runs of these methods in a new table in the Rebuttal PDF, and will update the main text with a full set of runs.

---

> > ### Comment · Reviewer_qXm9 · 2023-08-11
> >
> > I thank the authors for their detailed answers to my questions and additional experiments.
> >
> > I am not completely convinced that the predictor eigenspace still aligns under the cosine loss under the same framework as for instance Tian et al., since the timescale argument is only acceptable in the quite restrictive NTK framework; and that this timescale argument is precisely proven after having supposed that the eigenspaces are aligned, making the argument cyclical. Although, I can see an argument that the optimal linear reconstructor is still the PCA matrix be valid if it is proven.
> >
> > Thank you for the additional Figures, which help to understand the effect of IsoLoss. I agree that (a) showcases well the theoretical effect of IsoLoss, which is the (early) recruitment of small eigenvalues, with similar magnitudes for all during training. (Although there is a constant decrease in value, due to weight decay maybe considering 1.c.) This effect is a bit present in (b) for the Euclidian case but seems much more difficult to analyze. In this case, it seems the learning dynamic is very different from the theoretical one, which seems surprising. Values are stable after a few epochs and span a high range of values, contrary to the theoretical optimal value of 1 for all. Do the authors have a hypothesis on this phenomenon?
> >
> > I agree with the justification for EMA, but find it important to include it in the justification of the method. Thank you for the addition.
> > I am however very surprised by the eigenvalue dynamic without EMA (Fig 1c). Do the authors have possible explanations for this phenomenon? If the main effect of the EMA should be eigenvalue recruitment, I am having trouble understanding the stark difference. Would it still be possible to have additional figures for the SimSiam (i.e. BYOL with no EMA) loss like for the BYOL loss? It may shed light on this difference if SimSiam eigenvalues behave similarly.
> >
> > I acknowledge the authors' justification for weight decay and thank them for the additional experiment, although the results of having lower values with higher weight decay do not seem particularly surprising. We agree that looking into the link with the number of eigenvalues recruited seems an interesting continuation of this work.
> >
> > I thank the authors for their changes based on my remarks.
> >
> > The addition of SimSiam and DirectCopy is welcome; although I am surprised by the accuracy gap between DirectCopy and DirectPred; although may be different when it will be implemented.
> >
> > The loss of the accuracy improvement over BYOL/SimSiam is disappointing and weakens the claims, but the main point of the authors about eigenvalues recruitment, speed, and the EMA is still valid and interesting; and I still tend toward accepting.

---

> > > ### Author Response · Authors · 2023-08-14
> > >
> > > We thank the reviewer for their fast response and for their further suggestions.
> > >
> > >
> > > > *”.. timescale argument is precisely proven after having supposed that the eigenspaces are aligned, making the argument cyclical. Although, I can see an argument that the optimal linear reconstructor is still the PCA matrix be valid if it is proven.“*
> > >
> > >
> > > Point well taken. We will revisit empirically to what extent this alignment happens for a linear trainable predictor when using the cosine loss (cf. Tian et al.) and we will work toward a proof. Finally, we will add a comment on this point in the final manuscript where DirectPred is introduced.
> > >
> > >
> > >
> > >
> > > > *”This effect is a bit present in (b) for the Euclidian case but seems much more difficult to analyze. In this case, it seems the learning dynamic is very different from the theoretical one, which seems surprising. Values are stable after a few epochs and span a high range of values, contrary to the theoretical optimal value of 1 for all. Do the authors have a hypothesis on this phenomenon?”*
> > >
> > >
> > > We were also initially surprised to see this discrepancy for the Euclidean case for DirectPred/IsoLoss. We think it is a numerical effect that is amplified by the log scale for the shared y-axis. Since IsoLoss and DirectPred converge to a lower range of eigenvalues (between $10^{-2}$ and $1$ vs approx 10-100 for cosine) the same amount of *absolute* fluctuations due to the online estimate of the covariance matrix will appear amplified. This explanation is also consistent with the much higher fluctuations we observe in the case of IsoLoss without EMA (see also next point). We will look into this matter in more detail.
> > >
> > >
> > >
> > >
> > >
> > >
> > > >”*I am however very surprised by the eigenvalue dynamic without EMA (Fig 1c). Do the authors have possible explanations for this phenomenon? If the main effect of the EMA should be eigenvalue recruitment, I am having trouble understanding the stark difference.*”
> > >
> > >
> > > We think this shows precisely that the network without EMA struggles more to recruit small eigenvalues. The same network with DirectPred does not train at all (not shown), whereas IsoLoss does much better. As suggested, we also looked at the eigenvalue dynamics for SimSiam and they appear to be a mix between the BYOL and IsoLoss (without EMA). We will include this result in the final version of the article.
> > >
> > >
> > >
> > >
> > > >”*The addition of SimSiam and DirectCopy is welcome; although I am surprised by the accuracy gap between DirectCopy and DirectPred; although may be different when it will be implemented.*”
> > >
> > >
> > > Thanks for the comment. This gap may be explained by the fact that the reported accuracy values are from the original publication and correspond to 500 epochs (all our runs used 1000 epochs). We did not manage to implement a version of DirectCopy that performed better. We will revisit DirectCopy carefully in our revised framework, particularly the predictor regularization strategy used in DirectCopy, and run it for the same number of epochs. We expect the performance gap to reduce or disappear.

---

### Official Review · Reviewer_qE19 · 2023-07-08

**Soundness:** 3 good
**Presentation:** 2 fair
**Contribution:** 4 excellent
**Rating:** 7
**Confidence:** 3

**Summary:**

The learning dynamics of non-contrastive self-supervised learning is an important problem to understand how these methods avoid collapse without using negative samples. In this paper, the authors provide a rigorous analysis of this problem on a simple linear network with Gaussian inputs, especially the difference between Euclidean and cosine loss functions. It is demonstrated that both losses have implicit regularization effects on the variance of representation, and the role of predictor and stop-grad operations are thoroughly investigated. Based on these insights, the authors propose an isotropic loss to equalize the convergence rate and lead to a better performance in various settings.

**Strengths:**

The theoretical analysis in this paper is novel and clearly improves the understanding of how loss function, stop grad and projector affect the performance of non-contrastive self-supervise learning. Based on the mathematical understanding, the authors propose a new loss function that could beat the baseline method on different datasets, which illustrates the validity and power of the proposed theory in practice.

**Weaknesses:**

The major weakness of this paper is the strong assumptions that require the network to be linear and the input to be iid Gaussian. For real-world datasets and practical network structures, it is not clear if those insights are still valid.

**Questions:**

1. In Table 1, it is predicted that $L_{noPred}$ converges slower than $L$, however, in Figure 2 there is an eigenvalue in $L$ that converge much slower, there seems to be a discrepancy between theory and experiments.

2. In theorem 1 and 2, the dynamics are taken to be the expectation of original dynamics over the distribution of augmentation. It seems that after taking expectations, it is essentially equivalent to inputting the original images without augmentations, which may lose an essential part of contrastive learning.

3. In Theorem 2, it is assumed that the eigenvalues are of comparable magnitude, is this assumption required only on the initial value, or on the whole trajectory? In the latter case, it seems not appropriate to draw conclusions on the behavior of eigenvalues.

**Limitations:**

No certain limits.

---

> ### Author Rebuttal · Authors · 2023-08-09
>
> Many thanks for enabling a further discussion of the finer points of our theoretical analysis. Please find below our point-by-point response.
>
> > *“For real-world datasets and practical network structures, it is not clear if those insights are still valid.”*
>
> Thanks for the question. Our empirical results suggest that our theory makes accurate qualitative predictions that do apply to real world data and deep neural networks such as ResNet-18 (see Fig.3 and eigenvalue dynamics in Rebuttal Fig. 1a,b).
> In the proof for Theorem 1, we further make the point that the less restrictive assumption of input data distributed among a finite set of mutually near-orthogonal clusters would also suffice. We will move this generalizing argument to the main text.
>
> > *“In Table 1, it is predicted that $L_{noPred}$ converges slower than $L$, however, in Figure 2 there is an eigenvalue in $L$ that converge much slower, there seems to be a discrepancy between theory and experiments.”*
>
> Thanks for this feedback. Perhaps it was not clear from the figure, but the x axis for the plots of $L\_{noPred}$ and $L\_{noPred}^{euc}$ are on a larger scale than the other plots to appreciate the slower convergence. We realize that this may cause confusion easily. To make it more clear, we will use the same axis-range in the revised version of the manuscript and add the current longer-timescale plot in an inset.
>
> > *“In theorem 1 and 2, the dynamics are taken to be the expectation of original dynamics over the distribution of augmentation. It seems that after taking expectations, it is essentially equivalent to inputting the original images without augmentations, which may lose an essential part of contrastive learning.”*
>
> Thanks for raising this important point. Indeed after taking the expectation, we lose the error component arising from the different augmentations. The main objective of our analysis was to study the macroscopic learning dynamics underlying collapse and divergence. These dynamics are dominated by the eigenvalues of the representational covariance matrix, and thus can be studied after taking the expectation, as we have demonstrated. Our theory does not provide a complete description of the microscopic dynamics underlying representation learning. Such a theory would require a powerful statistical model of the augmentations and nonlinear transformations in deep neural networks which are difficult to come by. However, Reviewer qXm9 pointed out that Wang et. al (DirectCopy) have made some exciting headway in this direction and we hope to develop our approach further in future work by integrating similar insights.
>
>
> > *“In Theorem 2, it is assumed that the eigenvalues are of comparable magnitude, is this assumption required only on the initial value, or on the whole trajectory? In the latter case, it seems not appropriate to draw conclusions on the behavior of eigenvalues.”*
>
> Thank you for the question. The assumption is not required. This information is a bit hidden, but can be found in the Appendix with the proof of Theorem 2. Specifically, we show that, even from an initial configuration where one eigenvalue is much larger than the others, the dynamics result in all the eigenvalues (including the single large one) becoming more equal over learning, leading to the comparable magnitude configuration. This is further supported by our simulation results (Rebuttal Fig. 1a). We realize that we should have made this point clear in the statement of the Theorem, and we will add this sentence in our revised manuscript.

---

### Official Review · Reviewer_ZNsZ · 2023-07-09

**Soundness:** 2 fair
**Presentation:** 2 fair
**Contribution:** 2 fair
**Rating:** 5
**Confidence:** 3

**Summary:**

This work analyzes the learning dynamics of non-contrastive SSL approaches such as BYOL and Simsiam. Based on the proposed theory, the authors analyze the how the stop-grad and predictor module affect the learning dynamics. Importantly, the authors design a theoretically inspired loss and gain improvement on classification tasks.

**Strengths:**

1. This work provides solid analysis and discussion on the learning dynamics of non-contrastive SSL algorithm, which contributes to our understanding of the algorithm. Especially, they consider the widely used cosine similarity loss, while existing works only discuss Euclidean loss. They also propose an isotropic loss which is inspired by the theory.
2. The simulation experiments helps understanding the proposed theory.

**Weaknesses:**

1. Lack of basic introduction to the used techniques, e.g., the neural tangent kernal (NTK).

2. Lack of awareness of dividing the article into paragraphs. The preliminary, theoretical analysis and the simulation experiment are all mixed in chapter 2, which might be confusing for readers.

3. The accuracy for the baseline model (BYOL) is too low. In the well-known SSL repository, solo-learn [1], the top-1 accuracy of BYOL (R-18, 1000ep) is 92.58 on CIFAR-10, and 70.46 on CIFAR-100. However, in this paper the baseline is only 89.4 and 61.1. I am afraid that this experiment cannot justify the effectiveness of the proposed isotropic loss.

4. This paper mainly provides theory to understand how non-contrastive SSL algorithms work (4 pages), and the design of the isotropic loss is inspired by the theory (only half a page). I think the title of this paper is not accurate enough.

**Questions:**

1. What does "Gaussian i.i.d inputs" in line 133 refer to specifically? To what extent does this setting fit the real-world scenario?

Others:
Line 68: Replace z^{(1/2)} with z^{(1)}, z^{(2)} to avoid ambiguity
Line 146: typo: "lammata"
$\ell_{iso}$ first appears in Fig 2. and Tab 1., which is before where it is defined.

If the author solves my concerns, then I would be willing to raise the rating.

[1] https://github.com/vturrisi/solo-learn

**Limitations:**

See weaknesses

---

> ### Author Rebuttal · Authors · 2023-08-09
>
> Thanks for your feedback. It helped us significantly improve our numerical results. Please find below our point-by-point response.
>
> > *“Lack of basic introduction to the used techniques, e.g., the neural tangent kernel (NTK).”*
>
> We are sorry this was missing in the initial submission. We will restructure the paper and add a dedicated “Background” section with an introduction to the NTK.
>
> > *“The preliminary, theoretical analysis and the simulation experiment are all mixed in chapter 2, which might be confusing for readers.”*
>
> Thanks for pointing this out. We will restructure the paper to separate the background, theory, and experiments. Specifically, we will split Section 2 into a “Background” section containing introductions to both DirectPred and the NTK, a “Theoretical results” section, and an “Experimental validation” section. We believe the paper will be clearer after these changes.
>
> > *“However, in this paper the baseline is only 89.4 and 61.1. I am afraid that this experiment cannot justify the effectiveness of the proposed isotropic loss.”*
>
> Thanks for this comment and pointing us to the solo-learn repository. We reimplemented all our simulations within this framework. As it contains several optimized choices of hyperparameters and augmentations, we found that it significantly improved the performance of all baselines and our own method (Rebuttal Table 1). While most claims remain justified with these new improved performance values, we can no longer hold the claim that our method outperforms nonlinear predictor methods. We will therefore remove this claim from the abstract and the relevant sections in the paper. IsoLoss still serves as an empirical validation of our theory since it correctly predicts faster initial convergences than DirectPred (see Rebuttal Fig.1d,e). Since IsoLoss was only a small application of our theory in the first place, we agree that it will be best to edit the abstract and article to focus on the analysis following your suggestion.
>
> > *“I think the title of this paper is not accurate enough.”*
>
> Thanks for making this point. We agree. and suggest the following new title: “Uncovering implicit variance regularization mechanisms in non-contrastive SSL”.
>
> > *“What does "Gaussian i.i.d inputs" in line 133 refer to specifically? To what extent does this setting fit the real-world scenario?”*
>
> Apologies for not specifying this properly. We refer to high-dimensional data in which every input component is drawn from a standard normal distribution. In the proof for Theorem 1, we further made the argument that the less restrictive assumption of input data distributed among a finite set of mutually near-orthogonal clusters would also suffice. We will move this generalizing argument to the main text. While these assumptions are violated by real-world data, our numerical simulations show that our theory makes qualitatively accurate predictions on ResNet-18 (Rebuttal Fig. 1) despite the Gaussian input assumption.
>
> > *“Others: Line 68: Replace z^{(1/2)} with z^{(1)}, z^{(2)} to avoid ambiguity Line 146: typo: "lammata". $l_{iso}$ first appears in Fig 2. and Tab 1., which is before where it is defined.”*
>
> Thanks for reading carefully. We will fix the notation and typos in the revised version of the manuscript.

---

> > ### Comment · Reviewer_ZNsZ · 2023-08-11
> > **Good Rebuttal**
> >
> > The authors did a good job at the rebuttal. Most of my previous concerns are resolved. They reproduce the experiments to match SOTA performance, and adjust their claims accordingly. This could help deliver the right message to readers (as far as I know, there is still no linear predictor on par with nonlinear ones, and the authors' new results again confirm this observation). Second, the authors also show that their advantage over DirectPred in the new setting, which I thought, would be enough for an empirical justification. I will further go through other reviewers' comments. I would be happy to raise the score when the button is available.
> >
> > One thing that came to my mind is that the theory from a recent work [1] might also be related, because it could also explain the effectiveness of IsoLoss. As far as I see, in their framework, the IsoLoss serves as a high-pass filter (the filter function is 2-\lambda, roughly speaking), which often outperforms the low-pass one like DirectPred (Table 1 & 2 [1]). A discussion of their relationship would be helpful.
> >
> > [1] Zhuo et al. Towards a Unified Theoretical Understanding of Non-contrastive Learning via Rank Differential Mechanism. ICLR. 2023.

---

> > > ### Author Response · Authors · 2023-08-14
> > >
> > > We thank the reviewer for the prompt response and positive comments. We also thank the reviewer for pointing out an interesting work in this space. The theory presented by Zhou et. al. looks indeed very relevant. We had also been thinking about what kinds of filters applied to the eigenvalues could help during learning. Looking at IsoLoss in their framework, as the reviewer suggested, should provide further insights. It may also help us study the functioning of the EMA more rigorously, which is still not fully clear to us. We will study the paper in more detail, and include it in our discussion in the revised version.

---

> > > ### Author Response · Authors · 2023-08-21
> > >
> > > Thanks again for your input. Since the discussion period is ending, if you're still happy to raise your score, you can do so through the “edit” button at the top-right of the initial review.

---

> > > > ### Comment · Reviewer_ZNsZ · 2023-08-21
> > > > **Score updated**
> > > >
> > > > Thanks for your reminder.

---

### Author Rebuttal · Authors · 2023-08-09

We thank the reviewers for the detailed and insightful feedback, which allowed us to further improve the article. Importantly, we improved our numerical results. The comments further motivated us to reconsider the scope of our article by focusing more on the theory part.

A common concern raised by the reviewers was whether our theoretical analysis still applies to practical settings with real-world data and large architectures. We provide new plots of the eigenvalue dynamics for learning in ResNet-18 that show empirically that our analysis holds to a surprising extent (Rebuttal Fig. 1a,b).

One reviewer raised concerns about model performance compared to SOTA implementations in the solo-learn package. To address these concerns, we re-implemented our simulations with solo-learn to take advantage of their optimized hyperparameter choices. This change improved the overall performance of our method to 69.4% on CIFAR-100 and 92.2% on CIFAR-10 (Rebuttal Table 1).
We confirmed that our theory’s qualitative predictions about isotropy remain correct. Specifically, the new implementation also shows that IsoLoss works without EMA unlike DirectPred, which fails to learn without EMA. Further, the learning curves (see Rebuttal Fig. 1d,e) demonstrate IsoLoss’ faster initial convergence as predicted by our theory.
The new implementation also enabled DirectPred to reach the same performance level as IsoLoss. However, this final accuracy does not match the best performance of BYOL (92.6% on CIFAR-10 and 70.5% on CIFAR-100). Therefore, we cannot defend the claim that IsoLoss outperforms BYOL/nonlinear predictor methods.

To accommodate these new findings, we will moderate the claims in the abstract and relevant passages and focus on the advances in theoretical understanding of the learning dynamics. We will use IsoLoss to demonstrate the merits of the analysis and remove the claim of outperforming nonlinear predictor methods. Despite this change, we believe that by shining a light on the learning dynamics our approach may help design improvements for negative-sample-free SSL. Consequently, we also propose to change the paper title to “Uncovering implicit variance regularization mechanisms in non-contrastive SSL” to reflect the shifted focus toward understanding.

---

> ### Author Response · Authors · 2023-08-21
>
> Thanks once more to all the reviewers for their feedback.

---

### Decision · Program_Chairs · 2023-09-21

**Decision:**

Accept (poster)

**Comment:**

This paper provides a rigorous analysis of non-contrastive self-supervised learning on a simple linear network with Gaussian inputs, especially the difference between Euclidean and cosine loss functions. It is demonstrated that both losses have implicit regularization effects on the variance of representation, and the role of predictor and stop-grad operations are thoroughly investigated. Based on these insights, the authors propose an isotropic loss to equalize the convergence rate and lead to a better performance in various settings.

All reviewers believe this paper makes valid contributions. The AC agrees and recommends acceptance.